# Horizontal gene transfer of molecular weapons can reshape bacterial competition

Elisa T. Granato[1,2*], Jacob D. Palmer[3,4*], Christian Kirk[1], Connor Sharp[5], George Shillcock[1], Kevin R. Foster[1,2,6*]

1 Department of Biology, University of Oxford, Oxford, United Kingdom, 2 Department of Biochemistry, University of Oxford, Oxford, United Kingdom, 3 Department of Biological Sciences, Binghamton University, Binghamton, New York, United States of America, 4 Binghamton Biofilm Research Center, Binghamton University, Binghamton, New York, United States of America, 5 School of Biological Sciences, University of Reading, Reading, United Kingdom, 6 Sir William Dunn School of Pathology, University of Oxford, Oxford, United Kingdom

* elisa.granato@gmail.com (ETG); jpalmer10@binghamton.edu (JDP); kevin.foster@path.ox.ac.uk (KRF)

## Abstract

Bacteria commonly use molecular weaponry to kill or inhibit competitors. Genes encoding many weapons and their associated immunity mechanisms can be transmitted horizontally. These transfer events are striking because they appear to undermine bacterial weapons when given to competing strains. Here, we develop an ecological model of bacterial warfare to understand the impacts of horizontal gene transfer. Our model predicts that weapon gene transfer from an attacker to a target strain is possible, but will typically occur at a low rate such that transfer has a negligible impact on competition outcomes. We tested the model empirically using a transmissible plasmid encoding colicin E2, a potent antibacterial toxin produced by *Escherichia coli*. As predicted by the model, we find that toxin plasmid transfer is feasible during warfare, but the resulting transconjugants remain rare. However, exploring the model further reveals realistic conditions where transfer is predicted to have major impacts. Specifically, the model predicts that whenever competing strains have access to unique nutrients, transconjugants can proliferate and reach high abundances. In support of these predictions, short- and long-term experiments show that transconjugants can thrive when nutrient competition is relaxed. Our work shows how horizontal gene transfer can reshape bacterial warfare in a way that benefits a weapon gene and strains that receive it. Interestingly, we also find that there is little cost to a strain that transfers a weapon gene, which is expected to further enable the horizontal gene transfer of molecular weapons.

## Introduction

Bacteria inhabit a wide range of environments, often forming densely populated communities where they engage in competition over scarce nutrients and space

**Data availability statement:** All experimental datasets generated during this study are available on the Zenodo data repository (https://doi.org/10.5281/zenodo.10909492). Our source code and simulation outputs are available on the Zenodo data repository (https://doi.org/10.5281/zenodo.14910561).

**Funding:** ETG is funded by a BBSRC Discovery Fellowship (BB/V004328/1). KRF is supported by Wellcome Trust Investigator award 209397/Z/17/Z and European Research Council Grant 787932. The funders had no role in study design, data collection and analysis, decision to publish, or preparation of the manuscript.

**Competing interests:** The authors have declared that no competing interests exist.

**Abbreviations:** AHT, anhydrous tetracycline; BtuB, B twelve uptake;  protein BCFU, colony-forming unit; ColE2, colicin E2; DAP, diamino pimelic acid; OD, optical density; ODEs, ordinary differential equations; oriT, origin of transfer

[1–3]. In this context, bacteria have evolved a diverse array of molecular weapon systems to inhibit or kill competing microbes [4,5]. A striking feature of many bacterial weapons is that they are encoded on mobile genetic elements. There is evidence that genes encoding the production of bacteriocins, antibiotics, and even complex weapons like the contractile type VI secretion machinery, can be horizontally transmitted between cells via transformation [6], conjugation [7–15], or transduction [16,17]. In most cases, highly specific immunity genes are co-transmitted with toxin genes, thereby protecting the recipient cell from further attack by the toxin in question [6–8,16,18–20].

Producing anti-competitor toxins can confer large benefits to bacteria living in communities [21–27]. Conversely, being immune to a toxin produced by a competitor can make the difference between life and death for a targeted strain. The horizontal mobility of toxin and immunity genes, therefore, presents a unique problem. If gene transfer happens during an encounter with a competitor, the target can become immune to the toxin, rendering the bacterial weapon useless and negating any potential fitness benefits for the producer. The transfer of bacterial weapons thus has the potential to rapidly reshape bacterial communities, defining which strains survive and which are driven to extinction. Despite this, and the evidence that weapons can be transferred [6–17], the potential consequences of weapon gene transmission between bacterial competitors remain poorly understood.

Here, we investigate the potential impacts of weapon gene transfer during encounters between warring bacterial strains. We begin by developing ecological models of competition between toxin producers and sensitive strains, where the toxin-immunity gene pair is horizontally transmissible. These analyses predict that horizontal gene transfer can indeed enable survival of previously toxin-sensitive cells, but that the amount of transfer is drastically limited by the killing of potential recipient cells. We test our predictions empirically using competitions between *Escherichia coli* strains mediated by a small, transmissible plasmid encoding colicin E2 (ColE2), which is a potent nuclease toxin [28]. These experiments reveal that mobile toxin plasmids can indeed be transferred to competitors, but as predicted, the number of new transconjugants remains low. However, we find that introducing realistic metabolic diversity between strains can greatly amplify the ecological success of transconjugants, enabling them to invade a population of the original attacker strain and reach high abundance. In this way, weapon gene transfer can benefit the targets of bacterial warfare and mobile weapon genes, if not the bacteria that actually transfer the weapons.

## Results

### Modeling predicts that killing of target cells limits weapon gene transfer

We model competition between two strains in a patch with a single limiting nutrient, where the attacker strain harbors a plasmid which has genes for toxin production, toxin release, and immunity, but can also transfer this plasmid to the target strain in the patch. The target is initially susceptible to the toxin, but if it receives the plasmid, it becomes a new genotype: the transconjugant. The transconjugant now has the

PLOS Biology

genes for toxin production, toxin release, and immunity, but otherwise maintains the genotypic background of the initial target strain. These three strain types will be referred to as the attacker, target, and transconjugant.

We are interested in the impact of horizontal gene transfer on toxin-mediated competition between bacterial strains. To gain some intuition, we began by first modelling the population dynamics of a system characterized either by toxin-mediated competition, or by horizontal gene transfer, but not both. In a scenario where the attacker cannot transfer the toxin-producing plasmid, there are no transconjugants, and the target population is rapidly killed by the toxin, leaving the toxin-producing attacker as the lone surviving population (Fig 1a and 1d). However, this outcome is dependent on

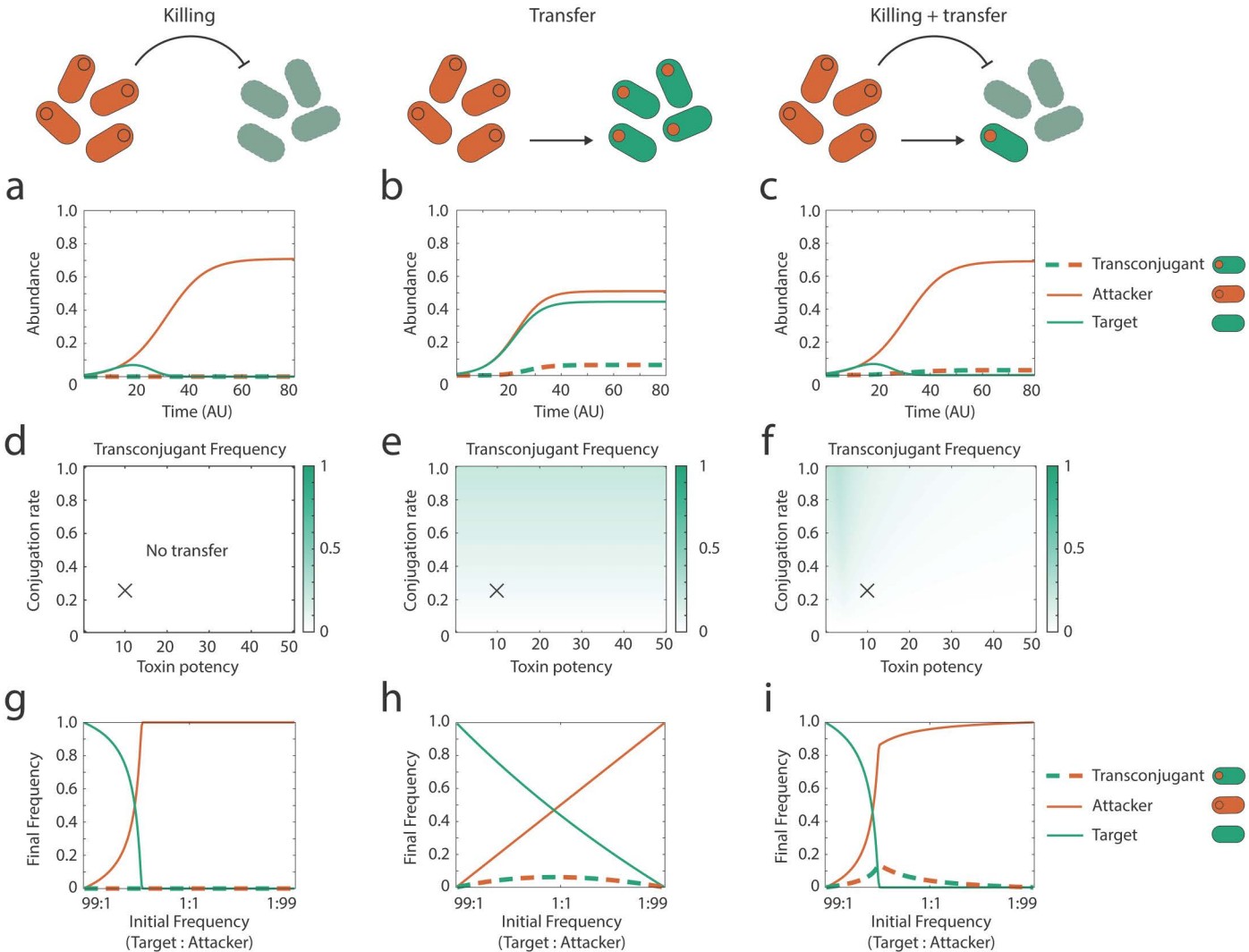

**Fig 1. Horizontal transfer of bacterial weapon genes is limited by the killing of potential recipients.** Modeling scenarios for each column are shown across the top row. **(a–c)** Example dynamics of the strains (attacker, target, transconjugant) during a contest using parameters that correspond to the cross (X) shown in the parameter sweeps directly below. **(d–f)** Transconjugant frequency at steady state (see section "Materials and methods") in competitions as a function of conjugation rate (*b*) and toxin killing efficiency (*E*). **(g–i)** Final frequency of different strain types at steady state as a function of the initial frequency of target and attacker strains. Top and middle rows are conducted at an initial frequency of 1:1:0 (target:attacker:transconjugants). All other parameters are default (Table 1) unless stated. Code and data underlying these figures are available from https://doi.org/10.5281/zenodo.14910561.

initial frequencies: if target cells greatly outnumber attackers to begin with, some targets can persist (Fig 1g). With only horizontal gene transfer and no toxin-mediated killing, transconjugants are generated as expected and all three genotypes (attacker, target and transconjugant) occur (Fig 1b).

We next considered both horizontal gene transfer and toxin-mediated competition in combination, which again results in transconjugants, but at a much lower frequency than the model that includes only HGT (Fig 1c). This result occurs because the killing of the target cells by the toxin greatly reduces the available cells for conjugation. Consistent with this, we see higher levels of transfer at low toxin potency (Fig 1f) and at relatively low numbers of attackers (Fig 1i), while the conjugation rate has a relatively small effect (Fig 1f). When attackers equal or outnumber the target cells in abundance, the amount of transfer is minimal as the targets are largely eliminated before transfer can occur (Fig 1i).

In sum, our model supports the basic intuition that the transfer of weapons can occur during bacterial warfare. However, all else being equal, the levels of transfer are typically lower than for typical horizontal gene transfer because, during warfare, many of the potential recipient cells will be killed off before transfer can happen. This effect is so strong that, if toxins are potent or attackers are common, there may be little or no transfer. While weapon gene transfer is feasible, it is not guaranteed, making it an empirical question whether it would actually occur in practice.

## Experiments demonstrate weapon gene transfer to competitors

To test the predictions of our model empirically, we turned to a well-characterized family of antibacterial toxins: the colicins. These potent protein toxins are produced by *Escherichia coli* and other *Enterobacteriaceae* in order to kill closely phylogenetically-related competitor strains and species [28]. In *E. coli*, colicins are exclusively encoded on plasmids, which also encode cognate immunity proteins protecting producer cells from their own toxins [28,29]. Strikingly, the majority of colicin plasmids are predicted to be transmissible between cells [28–33], and transfer has been observed in vivo [8], suggesting that horizontal transfer may indeed occur as part of their natural ecology.

Here we focus on the well-studied colicin E2, a nuclease toxin encoded on the pColE2-P9 plasmid (hereinafter: 'pColE2') originally isolated from *Shigella sonnei* [34]. The pColE2 plasmid does not encode its own conjugative transfer machinery, but is able to hijack transfer machinery produced by a conjugative plasmid residing in the same host strain [35,36]. To mobilize pColE2, we chose the broad-host-range conjugative plasmid R751 [37,38], since it was previously shown to mobilize the closely related colicin plasmid pColE1 [39,40]. For simplicity, our model assumes that cells are well-mixed in a given patch. However, our experiments introduce a degree of local spatial structure by studying bacteria growing on agar. As we will show, our modeling is robust to this introduction of spatial structure with predictions that are well supported by the experiments.

As in the modeling, we study three scenarios: toxin-based killing only, transfer only and then the two in combination. To study toxin-based killing only, we cocultured two genotypes of *E. coli* BZB1011: a toxin-producing strain (attacker; carrying pColE2), and a toxin-sensitive strain (target). Under these conditions, without the possibility of horizontal gene transfer, the target is rapidly killed (Fig 2a), leading to domination of the environment by the attacker. Despite the lack of horizontal transfer in this assay, a small number of target cells survive for several hours (Fig 2a). Colicin E2 (ColE2) relies on the outer membrane receptor BtuB for its killing activity [28,41], and mutations affecting this receptor are readily selected for in laboratory environments [42]. Consistent with this, ~93% (14/15) target clones sampled across all time points exhibit a multicolicin resistant phenotype (S1 Table), likely mediated by de novo mutations in *btuB* (see section "Materials and methods"). We also observe highly similar killing dynamics and rapid resistance evolution in a different strain background (MG1655; S1a Fig and S1 Table), confirming that ColE2 kills *E. coli* strains rapidly as long as the BtuB receptor is present in the targeted cells. We detect no transconjugants in experiments where conjugation is not expected, and transconjugant count data is therefore omitted from our plots (see section "Materials and methods").

To study transfer without toxin-based killing, we cultured a target strain that is resistant to ColE2 (BZB1011 Δ*btuB*), with an attacker that carries the colicin plasmid and the helper plasmid R751 (this attacker-donor genotype hereinafter referred

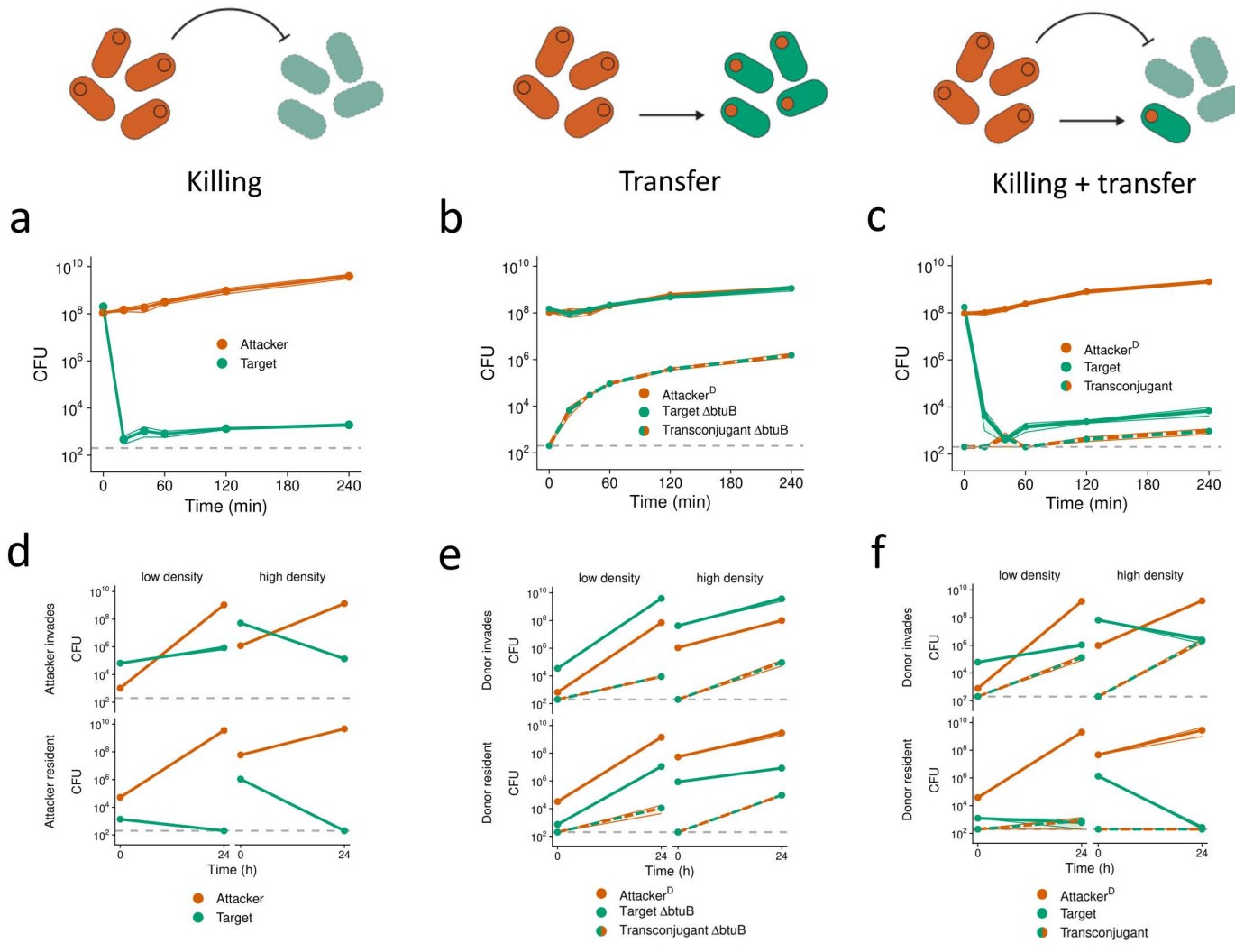

**Fig 2. Experiments show weapon gene transfer but with limited impact on strain frequencies.** We conducted pairwise competition assays between different *E. coli* strains on LB agar plates. For each genotype, cell recovery (CFU) at each time point of co-culturing is shown. Means across *n* = 3–6 biological replicates are shown as dots connected by lines. Shaded ribbons depict standard error across replicates. Dashed lines represent the detection limit (200 CFU). **(a–c)** Short competitions conducted over 4 h. CFU for each time point after *t* = 0 were determined via destructive sampling of replicates. (c) shows combined data from two independent experiments. **(d–f)** Long competitions conducted over 24 h at either low or high starting cell densities. **(a+d)** BZB1011-KmR pColE2-AmpR ('Attacker') competed against toxin-sensitive BZB1011-CmR ('Target'). No transconjugants (BZB1011-CmR pColE2-AmpR) were detected. **(b+e)** BZB1011-Km$^R$ R751-Sp$^R$ pColE2-Amp$^R$ ('Attacker$^D$') competed against toxin-resistant BZB1011-Cm$^R$ Δ*btuB* ('Target Δ*btuB*'). Transconjugant (BZB1011-Cm$^R$ Δ*btuB* R751-Sp$^R$ pColE2-AmpR) CFU are shown as they emerge during the interaction ('Transconjugant Δ*btuB*'). **(c+f)** BZB1011-Km$^R$ R751-Sp$^R$ pColE2-Amp$^R$ ('Attacker$^D$') competed against toxin-sensitive BZB1011-Cm$^R$ ('Target'). 'Transconjugant' (BZB1011-Cm$^R$ R751-Sp$^R$ pColE2-AmpR) CFU are shown as they emerge during the interaction. To test for differences in target survival between (**d**; top right) and (**f**; top right), we used a two-sided, two-sample *t* test on log-transformed CFU counts (*t* = −6.18, *df* = 4, *p* = 0.004). Data underlying these figures is available from https://doi.org/10.5281/zenodo.10909492.

to as "attacker$^D$"). The pColE2 plasmid is readily transferred into the toxin-resistant target (Fig 2b) and, as expected for a mobilizable plasmid, this transfer depends upon the presence of the origin of transfer (oriT) region in pColE2 (S1c Fig). However, the frequency of pColE2 transconjugants rapidly plateaus during the experiment (Fig 2b). This pattern is consistent with entry exclusion, whereby the spread of the helper plasmid limits the transfer of the mobilizable pColE2 plasmid

into the same cells [38,43]. In further support of this process, when both attacker[D] and target carry the helper plasmid from the beginning, very few pColE2 transconjugants are detected (S1d Fig).

Our data show that both killing and transfer can occur in isolation. However, the data also underline the high potency of colicin toxins, which the model shows can be a problem for weapon transfer because target cells may all succumb to the toxin before transfer can occur. To test whether transfer can occur during bacterial warfare, we completed an attacker[D] with the ColE2-sensitive target strain. These experiments reveal that transfer can occur even though sensitive targets were rapidly killed off, with thousands of transconjugants detected after four hours (Fig 2c), with similar dynamics in an alternative strain background (MG1655, S1b Fig). Even in scenarios where killing is rampant, therefore, some toxin-plasmid transfer can occur where the cells receiving the pColE2 plasmid become resistant to further attacks through the cognate immunity gene on the pColE2 plasmid.

The final density of target cells is higher than that of transconjugant cells, which suggests that resistance evolution may also occur when toxin plasmids are transferred. Screening postcompetition target clones confirmed that resistance evolution can indeed occur, with ~87% (26/30) being resistant in the BZB1011 background (S1 Table), although none (0/13) were found in the MG1655 background, which may indicate a lower probability of resistance arising in this strain background (S1b Fig and S1 Table). In competitions where the toxin plasmid is transferred into a sensitive strain, survivor clones thus appear to be mixtures of immune transconjugants and toxin-resistant mutants, and their relative frequencies depend on the strain background. Importantly, we rarely observe pColE2 transconjugants with a resistance phenotype (S1 Table), which is expected given they are already immune to the toxin once conjugation has occurred, and so natural selection for resistance mutations is not expected. Overall, these experiments confirm that horizontal transfer of the pColE2 plasmid can occur during highly antagonistic, toxin-mediated competition. However, as predicted by the model, the total abundance of the resulting transconjugants is very low, and they are outnumbered by the attacker $10^6$:1.

These first experiments were conducted at high cell densities, using a 1:1 initial ratio of strain abundances, and with a short competition duration of 4 h. Cell density, strain frequency and competition duration can affect both bacterial warfare [42,44,45] and conjugation rates [46,47]. To test the robustness of our findings in a broader range of competition scenarios, therefore, we conducted additional competitions where one strain is rare and the other common, which is intended to capture the case where one strain is invading the other. In addition, we varied initial cell density and extended the competition duration to 24 h to follow interactions for longer (Figs 2d–2f and S2).

In the killing-only cases (without transfer), target cells are rapidly killed, and the high cell density cases decrease target survival in particular (Figs 2d and S2a), which is expected as bacterial weapons are typically most effective at high cell density [44]. In the transfer-only cases (without killing), transconjugant frequencies increase with initial cell density, in line with conjugation being promoted by higher cell densities (Figs 2e and S2b). Moreover, in support of our modeling predictions (Fig 1h), transconjugants emerge at similar abundances for both invasion scenarios (Fig 2d, top versus bottom). By contrast, and again in support of our predictions (Fig 1i), the abundance of transconjugants is strongly limited at high attacker frequency once killing is introduced (Fig 2f, top versus bottom). Specifically, with both killing and transfer, we detect the most transconjugants at high target frequency, i.e., when the attacker invades into the target (Fig 2f, top; S2c Fig). At low target frequency, there are ~$10^4$-fold fewer transconjugants and also some target strain extinctions (Fig 2f, bottom; S2c Fig), which is consistent with the high attacker frequency leading to the elimination of target cells before transfer can occur. Moreover, even when transconjugants are seen in the experiments, the final abundances of the target cells show no change (Fig 2f and 2d, top left) or only a ~10-fold increase (Fig 2f and 2d, top right) relative to the equivalent scenario without transfer (Two-sample *t* test; $t = 6.18$, $p = 0.004$). This observation supports the key prediction of our modeling that toxin plasmid transconjugants can emerge during toxin-mediated competitions under some scenarios, but that the resulting ecological impacts are minor.

## Modeling predicts that metabolic diversity increases the impacts of weapon gene transfer

Both our theory and experiments, therefore, suggest that weapon transfer has a limited impact on the outcome of bacterial competitions. However, we reasoned that this outcome may depend on the degree of resource competition between strains. Up to this point, we have considered competitors that are isogenic strains, which differ only in their carriage of a toxin-encoding plasmid. In natural environments, competing bacterial strains will rarely have this level of overlap in their resource requirements due to differences in their genomes and resulting metabolism [48,49]. This reasoning led us to hypothesize that metabolic diversity may enable transconjugants to proliferate better than our models and experiments have so far predicted. To explore this hypothesis, we introduce metabolic diversity between strains in our model. In the new model, there is again a pool of nutrients that is available to all strains, creating resource-based competition. However, because of metabolic differences between the strains, each strain background now also has access to a 'private' nutrient that the other strain background cannot use. That is, one private nutrient is accessible only to the attacker, and another private nutrient is accessible to only the target strain and any transconjugants which may emerge during the interaction.

We again started by modeling the system first with only toxin-mediated killing, where the effects of metabolic diversity between the strains is minimal. The toxins impose a very strong selective force, and the attacker is the only strain which survives to the end of the competition (Fig 3a). The introduction of metabolic diversity into the transfer-only model results in relatively high levels of transfer across conditions (Fig 3b, 3e, and 3h). However, the strongest impacts of metabolic diversity are seen with both killing and weapon gene transfer. In particular, the parameter space where transconjugants survive is much broader across toxin potency (Fig 3f) and initial strain frequencies (Fig 3i) than without metabolic diversity (Fig 1f,i). Additionally, with both killing and weapon gene transfer, the frequency of transconjugants increases as metabolic diversity increases (S3a Fig), whereas variation in metabolic diversity has no impact on the frequency of strains in the scenarios with only killing or only weapon gene transfer (S3b and S3c Fig). In sum, the model supports the hypothesis that metabolic diversity between strains can increase the impacts of weapon transfer during bacterial competitions. The importance of metabolic diversity for our predictions is robust to parameter sweeps that vary the initial density and initial frequency of the attacker and target strains (S8 and S9 Figs).

## Metabolic diversity favors both transconjugants and resistance evolution

Our model predicts that access to a private nutrient can greatly increase the frequency of weapon-carrying transconjugants during bacterial warfare. To test this prediction, we engineered a strain of *E. coli* (Δ*srlAEB*) to make it unable to utilize sorbitol, a sugar alcohol that is commonly metabolized by both commensal and pathogenic *Enterobacteriaceae* [50–52]. In competitions on nutrient medium containing sorbitol, the wildtype ancestor uses sorbitol and successfully invades into a population of mutants deficient in sorbitol uptake (S4a Fig). Note that this experimental design is slightly different to our second model, because only one of the two strains has a private nutrient, rather than both as in our modeling. However, we show in the supplement that this change does not affect our modeling predictions (S5 Fig).

We performed the same set of competition assays as before (Fig 2d–2f) but now the attacker strain with the colicin plasmid (pColE2) cannot use sorbitol (Δ*srlAEB*), while the target strain can, and the two are competed on LB agar supplemented with sorbitol. Compared to competitions between isogenic strains (Fig 2d–2f), the target performs better across almost all scenarios tested (Figs 4a–4c and S4b–S4d). Moreover, when both killing and transfer can occur, the frequency of transconjugants is increased relative to our first experiments (Figs 2f and 4c; two-sample *t* test, *t* = 5.05, *p* = 0.007 (top left); Welch's *t* test, *t* = 20.07, *p* = 0.002 (top right)). Most strikingly, when the attacker is invading the target strain, the resulting transconjugants are now able to reach abundances equal to the attacker by the end of the competition (Fig 4c, top right). In line with our model's predictions, therefore, metabolic diversity can significantly increase transconjugant frequencies by boosting population growth.

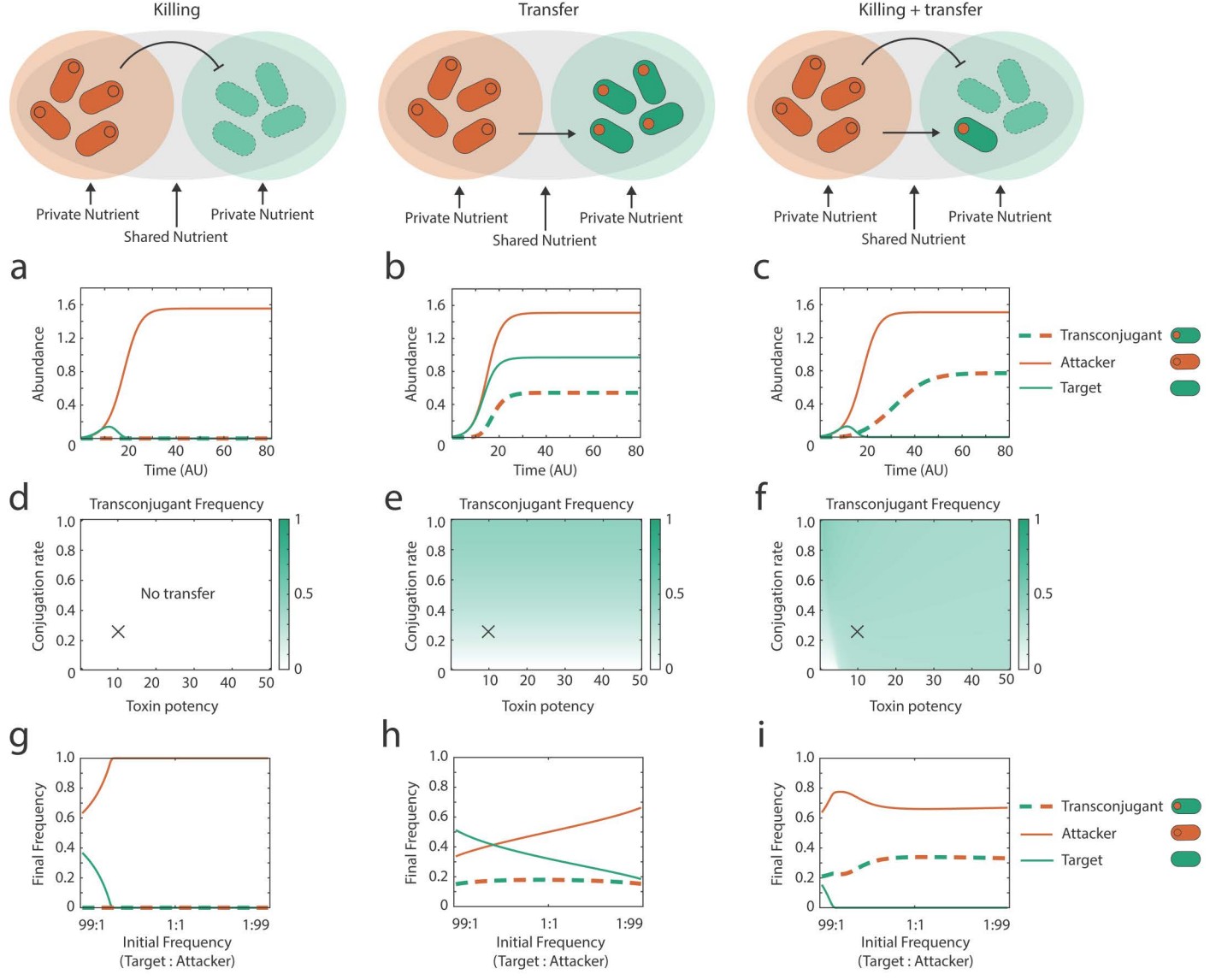

**Fig 3. Modeling predicts that metabolic diversity increases the ecological impacts of weapon gene transfer.** Modeling scenarios for each column are shown across the top row. **(a–c)** Example dynamics of the strains (attacker, target, transconjugant) during a contest using parameters that correspond to the cross (X) shown in the parameter sweeps directly below. **(d–f)** Transconjugant frequency at steady state (see section "Materials and methods") in competitions as a function of conjugation rate (b) and toxin killing efficiency (E). **(g–i)** Final frequency of different strain types at steady state as a function of initial frequency of target and attacker strains. Top and middle rows are conducted at an initial frequency of 1:1:0 (target:attacker:transconjugants). All other parameters are default (Table 1) unless stated. Code and data underlying these figures are available from https://doi.org/10.5281/zenodo.14910561.

However, one limitation of these competition assays is that they are relatively short, and the bacteria run out of limiting nutrients in the growth medium, at which point growth is arrested and fitness differences between strains cannot manifest anymore. To study longer-term strain dynamics, therefore, we modified our competition setup to include the periodic passaging of cells to fresh nutrient medium via replica-plating so that we could follow the competition for one week (Fig 4d and 4e). For these longer competitions, we restricted ourselves to the killing-only case and our focal case (killing

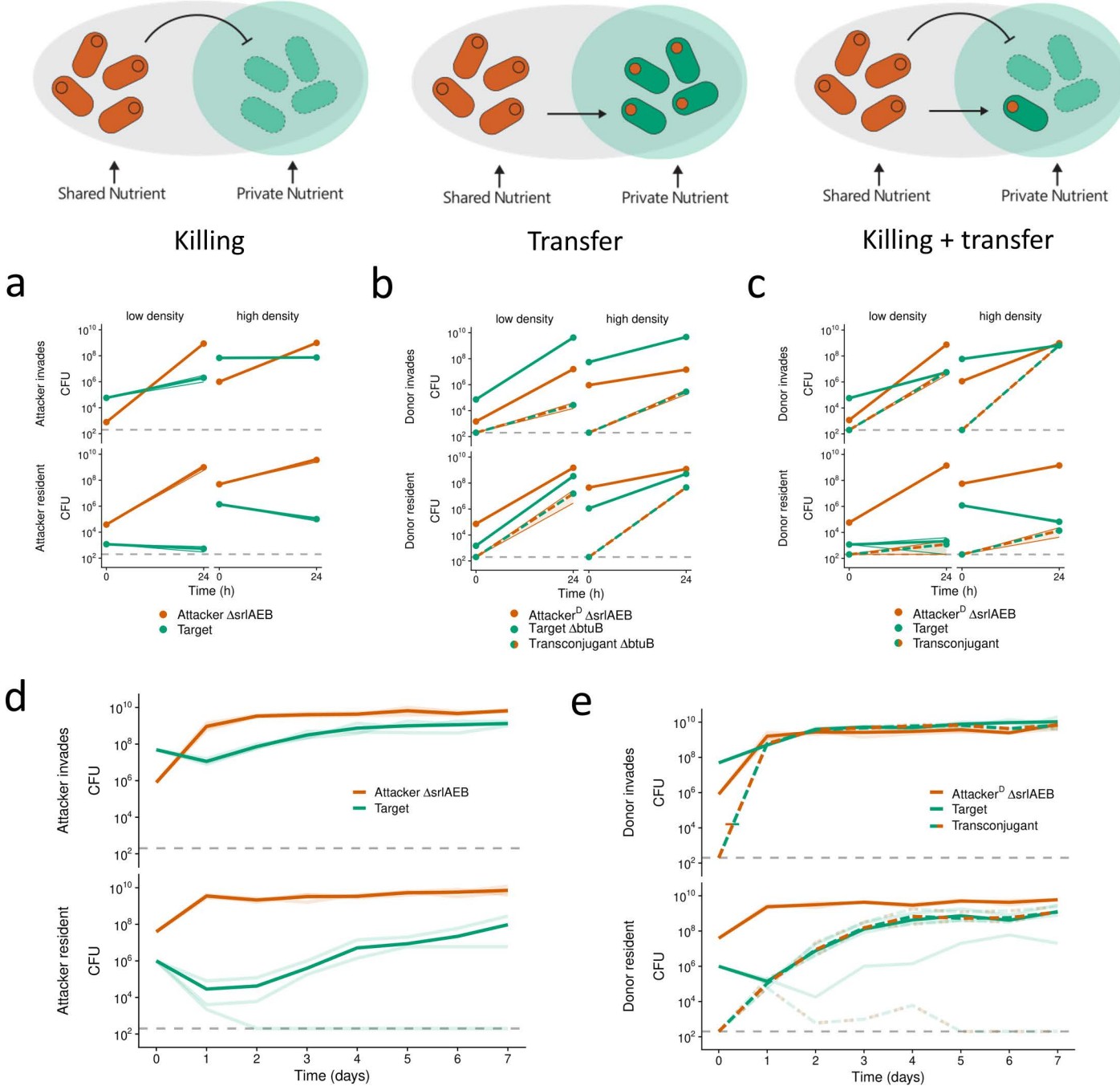

**Fig 4. A private nutrient promotes the invasion of weapon plasmid recipients, but also of spontaneously resistant cells.** We conducted pairwise competition assays between different *E. coli* strains on LB agar plates supplemented with sorbitol. For each genotype, cell recovery (CFU) at each time point of co-culturing is shown. **(a–c)** Competitions conducted over 24 h at either low or high starting cell densities. Means across *n* = 3 replicates are shown as dots connected by lines. Shaded ribbons depict standard error across replicates. **(d+e)** Serial passaging competitions conducted over 7 days. CFU were determined each day before transferring to fresh nutrient plates. Means across *n* = 3 replicates are connected by bold lines. Individual replicates are depicted as faint lines. **(a+d)** BZB1011-Km[R] Δ*srlAEB* pColE2-Amp[R] ('Attacker Δ*srlAEB*') competed against toxin-sensitive BZB1011-Cm[R] ('Target'). **(b)** BZB1011-Km[R] Δ*srlAEB* R751-Sp[R] pColE2-Amp[R] ('Attacker[D] Δ*srlAEB*') competed against toxin-resistant BZB1011-Cm[R] Δ*btuB* ('Target Δ*btuB*'). Transconjugant (BZB1011-Cm[R] Δ*btuB* R751-Sp[R] pColE2-AmpR) CFU are shown as they emerge during the interaction ('Transconjugant Δ*btuB*'). **(c+e)** BZB1011-Km[R] Δ*srlAEB* R751-Sp[R] pColE2-Amp[R] ('Attacker[D] Δ*srlAEB*') competed against toxin-sensitive BZB1011-Cm[R] ('Target'). 'Transconjugant' (BZB1011-Cm[R] R751-Sp[R] pColE2-AmpR) CFU are shown as they emerge during the interaction. To test for differences in final transconjugant

abundances between Figs 2f and 4c, we used two-sided, two-sample *t*-tests on log-transformed CFU counts: *t* = 5.05, *df* = 4, *p* = 0.007 (top left); *t* = 20.06, *df* = 2.02, *p* = 0.002 (top right; Welch's *t* test). Data underlying these figures is available from https://doi.org/10.5281/zenodo.10909492.

and transfer). With killing only (no transfer), the target drops in abundance during the first 24 h of competition, and in one replicate goes extinct (Fig 4d, bottom), but in the other replicates, the target then slowly recovers and increases in frequency over time. This long-term survival suggests resistance evolution is taking place during the experiment, as seen in the shorter competitions (Fig 2). This inference was confirmed by looking at the resistance profile of target clones isolated after 7 days of competitions across both scenarios (Fig 4d). 100% (15/15) of the target clones have a resistance profile consistent with a mutation in the receptor gene *btuB* (see section "Materials and methods"; S1 Table).

When we enabled transfer to occur alongside killing, transconjugants both emerged and reached high abundances in most cases (Fig 4e). Moreover, there was no evidence for resistance evolution whenever the transconjugants thrived, with all clones tested having a phenotypic profile consistent with being transconjugants rather than a general mechanism of resistance such as loss of BtuB (S1 Table). This observation suggests that they are exclusively immune to ColE2 due to carrying the immunity protein on pColE2, and resistance evolution did not occur in these clones. The exception to these patterns is one replicate where the target was invading the attacker (Fig 4e, bottom), here we observe resistance evolution (S1 Table) and do not detect the toxin plasmid in target isolates after day 4, suggesting that the plasmid was lost from the population over time.

Collectively, these results support our model's prediction that access to a private nutrient can empower transconjugants and promote their invasion into a population of toxin plasmid donors. However, in these experiments, when plasmid transfer is prevented, we find that target cells instead rapidly evolve resistance to the toxin and invade into toxin producers, again benefiting from access to private nutrients.

**Weapon gene transfer dominates competition outcomes when resistance evolution is costly**

Our models and experiments show that toxin plasmid transfer can enable strain survival in the face of a toxin-producing competitor. In our experiments, we also observed the rapid evolution of resistance against toxins, especially in scenarios without plasmid transfer. However, the ease with which this resistance occurs in the laboratory does not appear to be representative of resistance evolution in nature. Resistance against ColE2 in the laboratory is typically mediated by mutations in the gene encoding the outer-membrane receptor BtuB [28,41,42], preventing colicin import into the cell. However, BtuB ("B twelve uptake protein B") has an important function for *E. coli* and is necessary for import of vitamin B12 [53,54], a required co-factor for many metabolic enzymes, including those involved in essential amino acid synthesis [55]. Consistent with this, we find that BtuB is highly conserved in *E. coli*, with 2558/2601 genomes (98.4%) containing an intact *btuB* gene (see section "Materials and methods"). We therefore find no evidence that it is easily lost under natural conditions, but rather there appears to be strong natural selection to carry a functional BtuB receptor. Further consistent with this, when resistance to BtuB-targeting colicins in natural isolates has been found, it is mediated by BtuB-independent mechanisms, such as differences in LPS structure [56].

These patterns all suggest that the rise of BtuB mutants in our experiments is an artefact of the way that cells are grown in rich nutrient medium, where amino acid synthesis is not required for growth [57]. However, previous work has shown that it is possible to restore dependence on BtuB for growth by studying a strain background that is deficient for the vitamin-B12-independent methionine synthase MetE, involved in methionine biosynthesis under aerobic conditions [58,59]. This Δ*metE* strain requires a functional BtuB receptor to grow on minimal medium supplemented with vitamin B12 [59], mirroring the conserved nature of this receptor in natural isolates. We engineered a Δ*metE* strain and characterized its growth, survival and propensity to generate *btuB* mutants in short competitions with toxin producers (S6 and S7 Figs). In minimal medium supplemented with vitamin B12, Δ*metE* displays identical growth dynamics compared to the ancestral

WT both in liquid culture (S6a Fig) and on agar plates (S6b, S7a, and S7f Figs). Without B12 supplemented, the Δ*metE* strain grows very poorly, while the WT is unaffected, consistent with the new strain being reliant on BtuB (S6b Fig). More-over, when competed against a toxin-producing attacker, there is a much lower proportion of *btuB* mutants among the surviving clones (100% (8/8) for WT, ~38% (3/8) for Δ*metE*; S6c, S6d Fig and S1 Table).

These initial experiments support the notion that deletion of *metE* results in a high fitness cost of mutations in *btuB*, making *btuB* mutants less likely to be among the survivors of a competition. When we performed competitions with both toxins and transfer, pColE2 was readily transferred into Δ*metE* targets, with Δ*metE* transconjugants being present at slightly lower density after the competition than in the WT background (S6e Fig). We also confirmed that Δ*metE* can cap-italize on its growth advantage and invade into a population of Δ*srlAEB* cells when the minimal medium is supplemented with sorbitol (S7b Fig). We find that this effect is robust to variations in carbon source concentrations (S7c and S7d Fig) and to both strains carrying the helper and toxin plasmid (S7e Fig). Taken together, these results show that Δ*metE* is a highly suitable strain to study how the costs of resistance affect competition outcomes, since its phenotype is largely iden-tical to that of its WT ancestor except for a lower propensity of evolving *btuB*-mediated toxin resistance.

We therefore repeated the competitions as above (Fig 4), except using minimal media and where the target strain with access to the private nutrient sorbitol is either the WT or the Δ*metE* strain (Fig 5). In short competitions without plasmid transfer (Fig 5a), both target strains experience significant killing, with Δ*metE* targets surviving at slightly lower densities than WT targets when the attacker invades (two-sample $t$ test, $t = -9.36$, $p < 0.001$). This difference is consistent with a lower propensity for resistance evolution in the new Δ*metE* background. In further support of reduced resistance evolution in Δ*metE*, 3/3 of isolated WT survivors are putative *btuB* mutants versus 0/3 in the Δ*metE* background (Fig 5a, top; S1 Table). With plasmid transfer, pColE2 is readily transferred into both target strains, with lower final CFUs recovered for Δ*metE* compared to the WT (Fig 5b; two-sample $t$ test, $t = -7.57 | -8.83$, $p < 0.001$), and 0/3 putative *btuB* mutants among isolated Δ*metE* survivor clones (S1 Table). In the WT control 1/3 of isolated survivor clones are putative *btuB* mutants (S1 Table). These patterns then are all consistent with a cost to resistance evolution in the Δ*metE* strain, which makes resis-tance evolution via mutation less likely and instead favors immune transconjugants.

Finally, we conducted serial passaging competitions to monitor the long-term strain dynamics when resistance evolu-tion is costly. Without plasmid transfer, Δ*metE* targets drop rapidly in density within the first 24 h, even when they started at an initially high density (Fig 5c, top). In two replicate lines of this scenario, the Δ*metE* targets go extinct (day 3 and day 7, respectively). In the third replicate, the targets persist at low densities until the end of the competition. Isolated survivor clones ($n = 6$) from this replicate are resistant to several colicins, but also have a functional BtuB receptor, suggesting that resistance evolved via a different route (possibly TolB; see section "Materials and methods"). When Δ*metE* started at a lower density than the attacker (Fig 5c, bottom), it goes extinct in all three replicates by day 2. In scenarios with plasmid transfer, Δ*metE* targets are rapidly converted to transconjugants and increase in density over time (Fig 5d). Isolated survi-vor clones ($n = 6$) were all shown to be transconjugants with no evidence for resistance evolution (S1 Table).

In conclusion, these results show that, when resistance to toxins carries a fitness cost, toxin plasmid transfer can trans-form competition outcomes and represent the difference between extinction and competitive success of targeted strains.

## Discussion

Bacteria living in dense and diverse microbial communities are consistently faced with competition from other strains and species [1–3]. These communities are also hotspots for horizontal gene transfer, a key driving force for bacterial diversifi-cation and evolution [60–63]. Our modeling predicts that horizontal transfer of toxin and immunity genes can occur during bacterial warfare. The models also predict that the ecological impact of this transfer can be limited by a combination of toxin-mediated competition that eliminates potential recipients, and nutrient competition that limits the proliferation of any transconjugants that do arise (Fig 1). We see evidence for the potential importance of both of these competitive effects in our experiments with *E. coli* (Fig 2). However, further modeling suggested that introducing realistic metabolic diversity

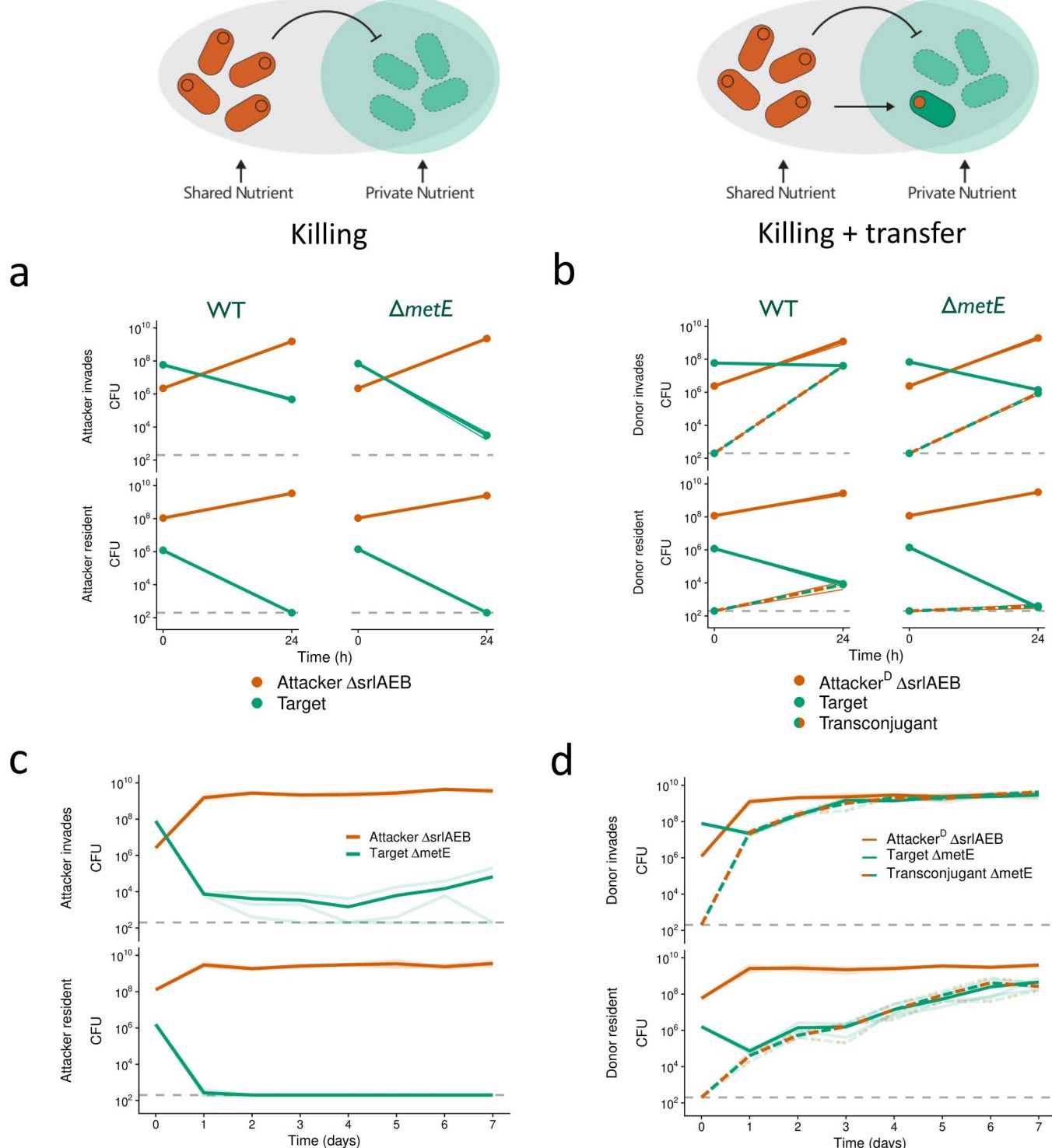

**Fig 5. Weapon gene transfer fundamentally alters competition outcomes when resistance is costly.** We conducted pairwise competition assays between different *E. coli* strains on minimal medium agar plates supplemented with glucose, sorbitol, and vitamin B12. For each genotype, cell recovery (CFU) at each time point of co-culturing is shown. Dashed grey lines represent the detection limit (200 CFU). **(a+b)** Competitions conducted over 24 h. Means across *n* = 3 replicates are shown as dots connected by lines. Shaded ribbons depict standard error across replicates. To test for

differences in target survival, we used two-sided, two-sample *t*-tests on log-transformed CFU counts. (a) WT, top left versus Δ*metE*, top right: t = −9.36, *df* = 4, *p* < 0.001. (b) WT, top left versus Δ*metE*, top right: *t* = −7.57, *df* = 4, *p* < 0.001. (b) WT, bottom left versus Δ*metE*, bottom right: *t* = −8.83, *df* = 4, *p* < 0.001. **(a)** BZB1011-Km^R Δ*srlAEB* pColE2-Amp^R ('Attacker Δ*srlAEB*') was competed against toxin-sensitive BZB1011-Cm^R or BZB1011-Cm^R Δ*metE* ('Target'). **(b)** BZB1011-Km^R Δ*srlAEB* R751-Sp^R pColE2-Amp^R ('Attacker^D Δ*srlAEB*') was competed against toxin-sensitive BZB1011-Cm^R or BZB1011-Cm^R Δ*metE* ('Target'). Transconjugant (BZB1011-Cm^R (Δ*metE*) R751-Sp^R pColE2-AmpR) CFU are shown as they emerge during the interaction ('Transconjugant'). **(c+d)** Serial passaging competitions conducted over 7 days. CFU were determined each day before transferring to fresh nutrient plates (see section "Materials and methods"). Means across *n* = 3 replicates are connected by bold lines. Individual replicates are depicted as faint lines. **(c)** BZB1011-Km^R Δ*srlAEB* pColE2-Amp^R ('Attacker Δ*srlAEB*') competed against toxin-sensitive BZB1011-Cm^R Δ*metE* ('Target Δ*metE*'). **(d)** BZB1011-Km^R Δ*srlAEB* R751-Sp^R pColE2-Amp^R ('Attacker^D Δ*srlAEB*') competed against toxin-sensitive BZB1011-Cm^R Δ*metE* ('Target Δ*metE*'). Transconjugant (BZB1011-Cm^R Δ*metE* R751-Sp^R pColE2-AmpR) CFU are shown as they emerge during the interaction ('Transconjugant Δ*metE*'). Data underlying these figures is available from https://doi.org/10.5281/zenodo.10909492.

between strains could enable the expansion of transconjugant populations (Fig 3). In support of this, experiments where transconjugants had access to a nutrient that the attacker could not use identified conditions where weapon gene transfer had major impacts on competition outcomes (Fig 4). These impacts were particularly strong when the evolution of de novo resistance to the weapon was costly (Fig 5). There, weapon gene transfer allowed the recipient strain to survive and establish where it would otherwise have gone locally extinct.

We have focused here in particular on the case of the colicin plasmids of *E. coli*. Our results show that colicin plasmid transfer can dramatically change competition outcomes, but in a manner that depends on ecological context. In healthy mammalian guts, *E. coli* and other *Enterobacteriaceae* typically only occur at very low densities [48,64–67], which is expected to impede plasmid-driven HGT since it requires direct cell-cell contact. More generally, in diverse communities of bacteria, there is the potential for 'social insulation' whereby third-party species reduce the potential for interactions between strains of a focal species [68,69]. However, during periods of inflammation, members of the *Enterobacteriaceae* can proliferate rapidly and reach high cell densities, promoting both HGT [8] and colicin-mediated competition [27]. These observations suggest that colicin plasmid transfer may have the greatest impacts on competition outcomes during these blooms.

Throughout our work, we find evidence that weapon gene transmission can benefit a target strain by enabling its survival, and sometimes also proliferation, in scenarios where it would otherwise go extinct. We did not find evidence that an attacker will benefit from transmitting its weapon. These findings raise an important question: why are many bacterial weapon genes transmissible if this primarily benefits the competitors of their current host? The answer may lie in natural selection on the genetic element that carries the weapon, as opposed to natural selection on the strain that carries the genetic element. This potential is particularly clear for weapons that reside on plasmids. In their relationship with bacterial hosts, plasmids can exist anywhere on a spectrum from mutualism to parasitism [70–72]. Moreover, this relationship can depend not only on the specific plasmid-host combination but also on the ecological context [73,74]. Consequently, the evolution of traits such as horizontal transmission can be shaped by how much they benefit the plasmids themselves, rather than their bacterial hosts [75–78].

Our work suggests that a weapon plasmid will benefit most from transmission whenever a target strain has a different nutrient niche to an attacker, because this allows the transconjugant to proliferate after transfer. There may also be a bet-hedging component if transfer increases the probability that a plasmid enters the host strain that is fittest in its local environment [70]. Consistent with the importance of transmission for colicins, colicin genes are not found in *E. coli* chromosomes [28,29], a common location for bacteriocin genes in other Proteobacteria [79,80]. The benefits of colicin production alone, therefore, might not be strong enough or frequent enough to favor colicin genes that are on the chromosome. Instead, frequent transmission may be necessary to support the lifestyle of mobile colicin plasmids [28–33], where plasmids that lose the ability to transfer are themselves lost over time. Consistent with this notion, sustained horizontal transmission has been found to play a pivotal role in the persistence of bacterial plasmids, including cases where natural selection for plasmid-encoded traits is weak or absent [77,78,81]. Interestingly, a different group of bacteriocins produced

by *E. coli* – the microcins – are also strongly associated with gene mobility, as many of them are encoded on plasmids [29,82], and those encoded on chromosomes are frequently associated with transposable elements [83–85]. This further supports the view that, in *Enterobacteriaceae,* genes encoding protein toxins commonly rely upon horizontal transmission.

The mobility of bacterial weapon genes, therefore, can be explained by benefits to the weapon genes themselves and any associated mobile genetic elements. But why do bacterial hosts seemingly permit the transfer of weapon genes to their competitors? Here, the answer may lie in the fact that the evolutionary costs to weapon gene transfer tend to be low for a host cell. In both the models and experiments, we find that the transconjugants that have received the weapon genes can only proliferate when there is weak nutrient competition between the strains (Fig 3–5). With strong nutrient competition, due to high metabolic overlap between the strains, transconjugants do not reach high abundances (Figs 1 and 2). When proliferation of transconjugants occurs, the associated weak nurient competition means that there is little or no impact on the abundance of attackers (e.g., compare Fig 5c and 5d). The result is that there may be little evolutionary benefit for a plasmid-carrying attacker to limit plasmid mobility, because it only leads to the outgrowth of transconjugants when they do not compete strongly with the donor for nutrients. Plasmid-borne weapon genes, on the other hand, can clearly benefit from horizontal mobility. In sum, while there may be strong natural selection on weapon genes to be mobile, natural selection on hosts to limit this mobility is likely to be much weaker, which may limit the evolution of host counter-adaptations.

Taken together, our results support the idea that mobile weapon genes can provide large benefits to the bacteria that carry them, because of the impacts of the weapons on competing strains. However, the mobility of the weapon genes does not appear to provide benefits to their host bacteria. Instead, those benefits fall on the weapon genes themselves and any associated mobile genetic element. Nevertheless, gene mobility can have important impacts on bacterial competition. When donor and recipient strains are limited by different nutrients, transfer can be followed by proliferation of the new weapon-carrying strain in the presence of the original attacker. The result is a starkly different ecological outcome to that without transfer, where the attacker would dominate, if not eliminate, the target strain. The horizontal transfer of bacterial weapon genes, therefore, has the potential to reshape bacterial warfare and the ecology of microbial communities.

## Materials and methods

### Modeling

**Differential equations – default model.** The core framework of the model is a set of ordinary differential equations (ODEs) based on established consumer-resource models incorporating toxin-mediated bacterial competition [45,86–88] and horizontal gene transfer [89–92].The model assumes the simplest case of conjugal plasmid transfer, where there are three potential bacterial populations: (1) attackers ($C_1$), which carry the toxin-bearing plasmid, and all of the components necessary for transfer, (2) targets ($C_2$), which lack the toxin-bearing plasmid, and (3) transconjugants ($C_3$), which are formerly members of the target population, but have acquired the toxin-bearing plasmid from an attacker, and can additionally transfer the toxin-bearing plasmid to targets. We use the model to follow the dynamics of these three bacterial populations ($C_1$, $C_2$, $C_3$), the toxins ($T$), which can be produced by both attackers and transconjugants, and a single consumable nutrient ($N_1$), used by all three bacterial populations.

$$\frac{dC_1}{dt} = C_1(t) * r_{C_1} * \frac{N_1(t)}{N_1(t) + K_{N1}} * (1 - \gamma) \tag{1}$$

$$\frac{dC_2}{dt} = C_2(t) * r_{C_2} * \frac{N_1(t)}{N_1(t) + K_{N1}} - b * C_2(t) * \left( C_1(t) * \frac{N_1(t)}{N_1(t) + K_{N1}} + C_3(t) * \frac{N_1(t)}{N_1(t) + K_{N1}} \right) - E * C_2(t) * \frac{T(t)}{T(t) + K_{C_2 T}} \tag{2}$$

$$\frac{dC_3}{dt} = C_3(t) * r_{C_3} * \frac{N_1(t)}{N_1(t) + K_{N1}} * (1 - \gamma) + b * C_2(t) * \left( C_1(t) * \frac{N_1(t)}{N_1(t) + K_{N1}} + C_3(t) * \frac{N_1(t)}{N_1(t) + K_{N1}} \right) \tag{3}$$

$$\frac{dT}{dt} = \gamma * C_1(t) * \frac{N_1(t)}{(N_1(t) + K_{N1})} + \gamma * C_3(t) * \frac{N_1(t)}{(N_1(t) + K_{N1})} - C_2(t) * \frac{T(t)}{(T(t) + K_{C_2 T})} \tag{4}$$

$$\frac{dN_1}{dt} = -\frac{N_1(t)}{(N_1(t) + K_{N1})} * (C_1(t) + C_2(t) + C_3(t)) \tag{5}$$

Within these equations, $b$ is the rate of horizontal transfer, $\gamma$ is the energy investment into toxin production, and $K_{N1}$ is the Monod saturation constant for $N_1$. We model the interaction between toxin and target species using the Hill equation, with a Hill coefficient of 1, as described previously [86]. Because the Hill coefficient is 1, the toxin-receptor interaction is functionally a Monod term, following similar dynamics as nutrients and the strains that consume them. We assume there is no inherent cost of carrying a plasmid or for conjugation, though conjugation is dependent on the availability of nutrients, as has been done previously [92]. A full list of variables and default parameters can be found in Table 1.

Experimental conditions for this study differ slightly from the modeling framework, where instead of a toxin-bearing plasmid capable of its own mobilization, the toxin-plasmid instead relies on mobilization via a 'helper' plasmid (see section "Strain construction"). While modeling this additional complexity has existing precedent [93], we are focused on the horizontal transfer of toxin-bearing elements between competing genotypes, rather than the population dynamics of multiple plasmids. We also experimentally observe extremely low levels of cells carrying only the toxin-bearing plasmid (see section "Selective plating"). Therefore, we argue that using the simplified form of horizontal transfer during toxin-mediated competition for modeling scenarios is sufficient.

**Differential equations – metabolic diversity model.** The core aspects of the default model (above) remain constant for the model incorporating metabolic diversity. We incorporate two private nutrients, where the first private nutrient ($N_2$) is available only to the attacker ($C_1$), while the second private nutrient ($N_3$) is available only to the target ($C_2$) and transconjugant ($C_3$) strains. Therefore, the different bacterial strains continue to consume and compete for the shared nutrient ($N_1$), but the private nutrients ($N_2$, $N_3$) simulate some degree of niche differentiation between the attackers and the targets and transconjugants. This is modelled by simply summing the Monod terms, as has been done previously

**Table 1. Default parameters.**

| Model parameter | Parameter description | Default values | Units |
|---|---|---|---|
| $C_1, C_2, C_3$ ($t = 0$) | Initial abundance (cell biomass) of each strain | 0.01, 0.01, 0 | g$C$ |
| $N_1$ ($t = 0$) | Initial pool of shared nutrient 1 | 1.0 | g$N$ |
| $N_2, N_3$ ($t = 0$) | Initial pool of private nutrients: nutrient 2, nutrient 3 | 1.0*, 1.0* | g$N$ |
| $T$ ($t = 0$) | Initial biomass of toxin | 0 | g$T$ |
| $K_{N1}, K_{N2}, K_{N3}$ | Saturation constant for nutrient uptake for each nutrient | 5, 5, 5 | g$N$ |
| $r_{C_1}, r_{C_2}, r_{C_3}$ | Maximum intrinsic growth rate | 1, 1, 1 | 1/$t$ |
| $E$ | Killing efficiency of toxin (i.e., potency) | 10 | 1/g$T$*$t$ |
| $b$ | Conjugation rate | 0.25 | 1/g$C$*$t$ |
| $\Gamma$ | Investment in toxin production | 0.15 | n/a |
| $K_{C_2 T}$ | Affinity of toxin for the target strain | 1.0 | g$T$ |

*$N_2$ and $N_3$ only apply to the model which includes metabolic diversity (Eqs. 6–12).

[86]. Note that equation 10 is the same as equation 5, above, but is still included in the second equation set for readability.

$$\frac{dC_1}{dt} = C_1\left(t\right) * r_{C_1} * \left(\frac{N_1\left(t\right)}{N_1\left(t\right) + K_{N1}} + \frac{N_2\left(t\right)}{N_2\left(t\right) + K_{N2}}\right) * \left(1 - \gamma\right)$$

(6)

$$\frac{dC_2}{dt} = C_2\left(t\right) * r_{C_2} * \left(\frac{N_1\left(t\right)}{N_1\left(t\right) + K_{N1}} + \frac{N_3\left(t\right)}{N_3\left(t\right) + K_{N3}}\right)$$
$$- b * C_2\left(t\right) * \left(C_1\left(t\right) * \left(\frac{N_1\left(t\right)}{N_1\left(t\right) + K_{N1}} + \frac{N_2\left(t\right)}{N_2\left(t\right) + K_{N2}}\right) + C_3\left(t\right) * \left(\frac{N_1\left(t\right)}{N_1\left(t\right) + K_{N1}} + \frac{N_3\left(t\right)}{N_3\left(t\right) + K_{N3}}\right)\right)$$
$$- E * C_2\left(t\right) * \frac{T\left(t\right)}{T\left(t\right) + K_{C_2 T}}$$

(7)

$$\frac{dC_3}{dt} = C_3\left(t\right) * r_{C_3} * \left(\frac{N_1\left(t\right)}{N_1\left(t\right) + K_{N1}} + \frac{N_3\left(t\right)}{N_3\left(t\right) + K_{N3}}\right) * \left(1 - \gamma\right)$$
$$+ b * C_2\left(t\right) * \left(C_1\left(t\right) * \left(\frac{N_1\left(t\right)}{N_1\left(t\right) + K_{N1}} + \frac{N_2\left(t\right)}{N_2\left(t\right) + K_{N2}}\right) + C_3\left(t\right) * \left(\frac{N_1\left(t\right)}{N_1\left(t\right) + K_{N1}} + \frac{N_3\left(t\right)}{N_3\left(t\right) + K_{N3}}\right)\right)$$

(8)

$$\frac{dT}{dt} = \gamma * C_1\left(t\right) * \left(\frac{N_1\left(t\right)}{N_1\left(t\right) + K_{N1}} + \frac{N_2\left(t\right)}{N_2\left(t\right) + K_{N2}}\right)$$
$$+ \gamma * C_3\left(t\right) * \left(\frac{N_1\left(t\right)}{N_1\left(t\right) + K_{N1}} + \frac{N_3\left(t\right)}{N_3\left(t\right) + K_{N3}}\right) - C_2\left(t\right) * \frac{T\left(t\right)}{T\left(t\right) + K_{C_2 T}}$$

(9)

$$\frac{dN_1}{dt} = -\frac{N_1\left(t\right)}{N_1\left(t\right) + K_{N1}} * \left(C_1\left(t\right) + C_2\left(t\right) + C_3\left(t\right)\right)$$

(10)

$$\frac{dN_2}{dt} = -\frac{N_2\left(t\right)}{N_2\left(t\right) + K_{N2}} * C_1\left(t\right)$$

(11)

$$\frac{dN_3}{dt} = -\frac{N_3\left(t\right)}{N_3\left(t\right) + K_{N3}} * \left(C_2\left(t\right) + C_3\left(t\right)\right)$$

(12)

**Differential equations – parameter sweeps.** In S8 Fig, we explore how initial abundance and initial frequency of attacker and target strains affect final frequencies of attacker, target and transconjugant strains, in both our initial (default) model as well as the model incorporating metabolic diversity. We find that initial abundance and frequency has limited impact on final transconjugant frequencies in either model. In contrast, the model incorporating metabolic diversity predicts much higher transconjugant frequencies than the model without metabolic diversity (compare S8a and S8b Fig). In S9 Fig, we explore how various parameters (toxin investment, initial abundance, toxin potency, conjugation rate, initial nutrients, and growth rate) affect final transconjugant frequencies in the two models with or without metabolic diversity.

## Experiments

**Bacterial growth conditions.** All strains were routinely grown overnight in 5 ml LB medium (10 g/L tryptone, 10 g/L NaCl, 5 g/L yeast extract) in 15 mL polypropylene tubes at 37 °C with agitation (220 rpm). Routine culturing on plates was carried out on 1.5% w/v LB agar. Where appropriate, the medium was supplemented with kanamycin (50 μg/mL), carbenicillin (100 μg/mL), spectinomycin (50 μg/mL), chloramphenicol (25 μg/mL) or gentamicin (20 μg/mL). Optical

density (OD) of liquid cultures was measured at 600 nm. For culturing of strain JKe201, LB medium was supplemented with 100 µM of diamino pimelic acid (DAP). For experiments involving LB supplemented with sorbitol (LB-Srb), 8% m/v sorbitol was mixed with LB medium at a 1:1 ratio, resulting in final concentrations of ½ LB and 4% w/v sorbitol. For experiments in minimal medium, bacteria were grown in M9 minimal medium (M9; 61.72 g/L $Na_2HPO_4$, 30 g/L $KH_2PO_4$, 10 g/L $NH_4Cl$, 5 g/L NaCl, 2 mM $MgSO_4$, 0.1 mM $CaCl_2$) supplemented with 0.4% w/v glucose (M9-Glc). For experiments involving M9 supplemented with sorbitol (M9-Glc-Srb), glucose and sorbitol were added to M9 medium at final concentrations of 0.2% w/v (glucose) and 4% w/v (sorbitol) unless indicated otherwise. For experiments involving M9-Glc or M9-Glc-Srb supplemented with vitamin B12 (M9-Glc-B12; M9-Glc-Srb-B12), vitamin B12 was added to M9-Glc or M9-Glc-Srb medium at final concentrations of either 1 nM (Figs 5a, 5b, S6 and S7) or 10 nM (Fig 5c and 5d). For all experiments conducted on agar plates, liquid medium was supplemented with 1.5% w/v agar. All experiments were conducted at 37 °C unless indicated otherwise.

**Plasmid construction.** To generate pColE2-Amp^R, the Amp^R fragment was amplified from pUC19 using primers GibsColE2_Amp_For and GibsColE2_Amp_Rev, and the pColE2-P9 vector was amplified using primers Out_pColE2_For and Out_pColE2_Rev. Fragment and vector were then joined using Gibson Assembly Master Mix (New England Biolabs) according to the manufacturer's instructions, and chemically competent BZB1011 cells were transformed with 0.5 µL of the reaction product. Transformants were selected on carbenicillin, and plasmids were isolated from candidate clones using QIAprep Spin Miniprep Kit (QIAGEN). The E2 colicin operon was sequenced and found to be free of mutations.

To construct pColE2-ΔoriT-Amp^R, the Amp^R fragment was amplified from pUC19 using primers amp_fw and amp_rv, and the pColE2-P9 vector was amplified without its oriT region using primers e2_amp_fw and e2_amp_rv. Fragment and vector were then joined using Gibson Assembly as described above, and chemically competent One Shot TOP10 cells (Invitrogen) were transformed with 0.5 µL of the reaction product. Transformants were selected on carbenicillin, and plasmids were isolated from candidate clones using QIAprep Spin Miniprep Kit (QIAGEN). Plasmid sequence and deletion of oriT was confirmed via sequencing using primers e2_1–7, amp_fw and ampR_fw.

All plasmids used in this study are listed in S2 Table. All primers used in this study are listed in S3 Table.

**Strain construction.** To generate BZB1011 Δ*metE* Cm^R, the *metE* gene was replaced with a chloramphenicol resistance cassette following the λ Red method outlined in [94]. PCR product was generated using primers metE_del_fw and metE_del_rv with pKD3 as the template, gel-purified, and eluted in nuclease-free water. The parent strain BZB1011 was transformed with plasmid pKD46 using electroporation and selection on carbenicillin, and BZB1011 pKD46 cells were cultured overnight in LB at 30 °C. Following subculturing in fresh LB for 2 h at 30 °C, cells were induced with 10 mM arabinose for 1 h at 30 °C. Cells were then made electrocompetent by washing three times with ice-cold Milli-Q water and concentrating 100-fold. Electroporation was performed using 50 µL of cells and 200 ng of PCR product in 0.1-cm electroporation cuvettes (Bio-Rad). Shocked cells were added to 1 mL LB, incubated for 2 h at 37 °C, and insertion mutants were selected on chloramphenicol at 37 °C. Candidate clones were screened using colony PCR (primers metE_ver_up and metE_ver_dw), and tested for sensitivity to carbenicillin. Selected clones were confirmed via sequencing of the *metE* region (primers metE_ver_up and metE_ver_dw). BZB1011 Km^R and BZB1011 Cm^R were constructed using the same λ Red method, except kanamycin and chloramphenicol resistance cassettes were introduced at a neutral locus between genes *yidX* and *yidA* [95]. PCR products were generated using primers yidX-yidA_CmKan_For and yidX-yidA_CmKan_Rev with templates pKD3 (CmR) and pKD4 (Km^R). Selected clones were confirmed via sequencing of the *yidX-yidA* region (primers yidX-yidA_ver_up and yidX-yidA_ver_dw). BZB1011 *srlAEB*::Km^R was constructed using the same λ Red method, except the kanamycin resistance cassette was introduced at the *srlAEB* locus [96]. PCR products were generated using primers srlAEB_del_for and srlAEB_del_Rev on template pKD4. Selected clones were confirmed via sequencing of the *srlAEB* region (primers srlAEB_ver_up and srlAEB_ver_dw).

For the generation of BZB1011 Δ*btuB* Cm^R, *btuB* was deleted from BZB1011 Cm^R following the method outlined in [97,98]. Briefly, JKe201 cells carrying pTML8 were mated with BZB1011 Cm^R for 6 h. Transconjugants were selected

on kanamycin, and three colonies were combined and cultured for 4 h in 2 mL LB. Deletion mutants were then selected on no-salt LB agar supplemented with 20% m/v sucrose and 0.5 μg/mL anhydrous tetracycline (AHT), and sequence-confirmed using primers TML-P9 and TML-P10.

For the generation of MG1655 Km$^R$, tn7-insertion of a kanamycin resistance cassette was performed following the method outlined in [99]. Briefly, JKe201 cells were transformed with pUC18R6KT-mini-Tn7T-Km via electroporation and selected on carbenicillin and DAP. Triparental mating was then performed for 5 h using strains MG1655 (recipient), JKe201 pUC18R6KT-mini-Tn7T-Km (donor), and JKe201 pTNS2 (helper). MG1655 insertion mutants were selected on kanamycin and confirmed to be carbenicillin-sensitive.

To generate strains carrying R751-Sp$^R$ (S2 Table), a general-purpose donor strain was created by transforming chemically competent One Shot TOP10 cells (Invitrogen) with R751-Sp$^R$ according to the manufacturer's instructions. Transformants were selected on spectinomycin. TOP10 R751-Sp$^R$ was then used to conjugate R751-Sp$^R$ into various recipient strains by mating on LB for 4 h, followed by selection on spectinomycin and appropriate antibiotics selecting against the TOP10 R751-Sp$^R$ donor.

To generate strains carrying pColE2-Amp$^R$ (S2 Table), a general-purpose donor strain was created by transforming chemically competent JKe201 with pColE2-Amp$^R$ and selecting transformants on carbenicillin and DAP. JKe201 pColE2-Amp$^R$ was then used to conjugate pColE2-Amp$^R$ into various recipient strains by mating on LB supplemented with DAP for 4 h, followed by selection on carbenicillin without DAP supplementation.

To generate strains carrying pColE2-ΔoriT-Amp$^R$ or pColE2-Cm$^R$ (S2 Table), recipient strains were made electrocompetent by washing three times with ice-cold Milli-Q water and concentrating 100-fold. Electroporation was then performed using 50 μL of cells and 200 ng of plasmid in 0.1-cm electroporation cuvettes (Bio-Rad). Shocked cells were added to 1 mL LB, incubated for 2 h, and transformants were selected on carbenicillin (pColE2-ΔoriT-Amp$^R$) or chloramphenicol (pColE2-Cm$^R$).

All strains used in this study are listed in S2 Table. All primers used in this study are listed in S3 Table.

**Competition assays.** To quantify growth in the presence of a competitor, competition assays were carried out on nutrient agar plates.

**Short competitions.** For short competitions at high cell densities (Figs 2a–c and S1), liquid precultures in LB were inoculated from glycerol stocks, supplemented with appropriate antibiotics selecting for chromosomes and any plasmids (S2 Table), and incubated overnight. Overnight cultures were washed 3 times with LB medium, diluted 1:100 into 5 mL of fresh LB medium supplemented with antibiotics selecting for any plasmids (S2 Table), and grown for 3 h with agitation. For each strain, 1 mL of preculture was then washed 5 times with 1 mL of saline (0.85% w/v), and resuspended and adjusted to an optical density at 600 nm (OD) of 4.0 using saline (0.85% w/v). Pre-competition colony-forming unit (CFU) counts were determined by serially diluting all density-adjusted monocultures and spotting 5 μL per dilution on selective LB agar plates. Strains were then mixed at a ratio of 1:1 (e.g., 500 μL:500 μL), and for each time point to be measured, mixed cell suspensions were spotted in triplicate 100 μL droplets on LB agar plates. Droplets were left to dry at room temperature for approximately 30 min before incubating the plates at 37 °C for a maximum duration of 4 h. After incubation, competition spots were washed off with 1 mL saline (0.85% w/v) each and subjected to seven rounds of 10-fold serial dilution. Post-competition CFU counts for each strain were determined by plating on LB agar supplemented with appropriate antibiotics (see section "Selective plating"; S2 Table) and colony counting after incubating either for 18 h at 37 °C or for 24 h at 30 °C.

**Long competitions at varying densities.** For long competitions at varying densities (Figs 2d–f, 4a–4c and S4d), liquid precultures in LB were inoculated from glycerol stocks, supplemented with appropriate antibiotics selecting for chromosomes and any plasmids (S2 Table), and incubated overnight. For each strain, 1 mL of preculture was then washed 5 times with LB medium, and resuspended and adjusted to OD = 2.0 using LB medium. Normalized cell suspensions were diluted 10-fold (OD = 0.2), 100-fold (OD = 0.02) or 1000-fold (OD = 0.002) to generate a range of cell densities. Pre-competition CFU counts were determined by serially diluting all diluted monocultures and plating

100 μL per dilution on selective LB agar plates. Competitor strain dilutions were then mixed at a ratio of 1:50 (e.g., 10 μL:490 μL) to create asymmetric initial frequencies. Mixed cell suspensions were spotted in triplicate 50 μL droplets on agar plates and left to dry at room temperature for approximately 30 min, before incubating the plates for 24 h at 37°C. For each experiment, the nutrient medium used for the agar plates is specified in the corresponding figure legend. After incubation, post-competition CFU counts for each strain were determined as outlined above (see section "Short competitions").

**Long competitions in minimal medium.** For long competitions on minimal medium agar, liquid precultures in M9-Glc-B12 were inoculated from single colonies grown on LB plates, supplemented with appropriate antibiotics selecting for chromosomes and any plasmids (S2 Table), and incubated overnight. For each strain, 1 mL of preculture was then washed either 3 times (S7b–d Fig) or 5 times (Figs 5a, 5b, S6c–e and S7e) with 1 mL of saline (0.85% w/v), and resuspended and adjusted to OD = 2.0 using saline (0.85% w/v). Pre-competition CFU counts were determined by serially diluting all density-adjusted monocultures and spotting 5 μL per dilution on selective LB agar plates. Competitor strain dilutions were then mixed at a ratio of 1:50 (e.g., 10 μL:490 μL) to create asymmetric initial frequencies. Mixed cell suspensions were spotted in triplicate 50 μL droplets on agar plates and left to dry at room temperature for approximately 30 min, before incubating the plates for 24 h at 37 °C. For each experiment, the nutrient medium used for the agar plates is specified in the corresponding figure legend. After incubation, post-competition CFU counts for each strain were determined as outlined above (see section "Short competitions").

**Serial passaging.** To observe more long-term interactions between strains, we conducted serial passaging experiments. Competitions were set up on agar plates as outlined above (see section "Long competitions at varying densities") using an initial cell density of OD = 2.0, and incubated for 24 h. Competition spots were then replica-plated onto fresh agar plates using sterile velveteen squares mounted on a plastic block. After replica plating, individual competition spots were each resuspended in 1 mL saline (0.85% w/v), serially diluted, and spotted on selective plates to determine CFU counts as outlined above (see section "Short competitions"). The newly inoculated plates were incubated for 24 h. For the experiments shown in Figs 4e and 5c, this was repeated a total of 6 times, resulting in a total competition duration of 7 × 24 h. For the experiment shown in S7f Fig, this was repeated twice in total, resulting in a total competition duration of 3 × 24 h. On the last day, competition spots were resuspended, serially diluted, and spotted on selective plates without prior replica-plating.

**Selective plating.** For all competition assays, we used selective plating to determine initial and final cell densities for each genotype. All relevant chromosomes and plasmids carried an antibiotic resistance marker (S2 Table). To determine CFU counts for initially plasmid-carrying (e.g., attacker or attacker-donor/attacker$^D$) strains, we used plates selecting for their respective chromosomes and all plasmids they harbored. To determine CFU counts for initially plasmid-free (target) strains, we used plates selective for their chromosomes.

For experiments in the BZB1011 strain background involving transfer of pColE2 by the helper plasmid R751-Sp$^R$, we determined post-competition densities of transconjugants by spotting on plates containing chloramphenicol (Cm; selecting for target strain chromosome) and carbenicillin (Cr; selecting for pColE2-Amp$^R$), as well as on plates containing Cm, Cr and spectinomycin (Sp; selection for R751-Sp$^R$). Because colony counts on both types of plates were nearly identical throughout all experiments – i.e., nearly all toxin plasmids transconjugants also received the helper plasmid – we used colony counts on Cm-Sp-Cr plates to represent transconjugant CFUs in all relevant figures. The same technique was applied to experiments in the MG1,655 strain background, albeit with different resistance markers (S2 Table).

For experiments where no pColE2 transfer was expected, i.e., experiments not involving the helper plasmid R751-Sp$^R$, we detected no colonies on plates selecting for pColE2 transconjugants after both short (4 h; Figs 2a and S1a) and long (24 h; Figs 2d, 4a, S2a and S4b) competition assays. Transconjugant counts are therefore not shown in these plots, and we did not include selective plates for pColE2 transconjugants for no-transfer regimes in subsequent experiments (Figs 4d, 5a, 5c, S6c and S6d).

**Resistance phenotyping.** To test whether target strains had spontaneously evolved resistance to colicin E2, we conducted post-competition resistance phenotyping assays (S1 Table). For experiments shown in Figs 1a, 1c, S1a and S1b, we sampled clones at every time point (20 min, 40 min, 60 min, 120 min, 240 min). For experiments shown in Figs 4d, 4e, 5a–5d and S6c, we only sampled clones at the final time point. In replicates where targets had become extinct before the final time point (e.g., Fig 5a), no clones were sampled. Colonies to be tested were picked from selective plates using a toothpick, inoculated into 200 μL LB in a 96-well plate, and incubated overnight. Per replicate, we aimed to pick one colony from each type of selective plate, selecting for either (i) the target strain chromosome, (ii) the target strain chromosome and the helper plasmid; (iii) the target strain chromosome and the toxin plasmid; or (iv) the target strain chromosome, the helper-, and the toxin plasmid (see section "Selective plating"). In parallel, colicin producers (colicin E2, E8, E1 and/or N) as well a non-producing WT negative control were cultured in 5 mL LB medium overnight. The next day, test clone cultures were diluted $10^{-3}$ into fresh LB medium, and 15–20 μL of diluted culture was spread on LB agar. Up to six clones were spread separately on a single LB agar plate. For each producer culture, 1 mL of supernatant was then harvested by centrifugation, filter-sterilized (0.2 μm), and 5–10 μL spotted on areas where targets were seeded. Droplets were dried at room temperature, and plates were then incubated for 24 h at 30 °C. For experiments shown in Figs 1a, 1c, 4d, 4e, S1a and S1b, test clones were exposed to colicin E2 and E8. For experiments shown in Figs 5a–5d and S6c, test clones were exposed to colicin E2, E8, E1 and N. Halos were considered evidence of killing by the colicin present in the supernatant. The *btuB* deletion mutant was used as a positive control for resistance against colicin E2 and E8. Colicin mono-producers were used as positive controls for immunity against their respective colicins. Depending on what colicins target clones were sensitive to, they were categorized as either "WT/sensitive", "E2 immune", "BtuB resistant", or "speculative TolB resistant" (Table 2). When targets were sensitive to all tested colicins, they were considered "WT". When target clones were able to grow when exposed to colicin E2, but were sensitive to colicin E8, they were categorized as "E2 immune". This phenotype is expected for all pColE2 plasmid carriers, including transconjugants. When targets were able to grow when exposed to both colicin E2 and E8, they were considered (multi-) colicin resistant, likely via mutation of the gene encoding the toxin-binding outer membrane receptor BtuB [28,41,42]. To confirm this, we Sanger sequenced (primers BtuB_BZB.F and BtuB_BZB.R) the *btuB* region of eight such putative "BtuB resistant" clones that emerged during a pilot competition experiment (S6c Fig). We found that 100% (8/8) of resistant clones in the WT background had large regions of *btuB* mutated by either large insertions or frame-shift mutations, consistent with results in previous studies [42]. In experiments using the BZB1011 Δ*metE* strain, mutations in *btuB* were predicted to incur a large fitness cost. Survivor clones from a subset of these competitions (Figs 5a–5d and S6c) were therefore subjected to additional resistance testing. Clones able to grow when exposed to colicins E2, E8 and E1, but sensitive to colicin N, were considered "BtuB resistant" since colicin E1 requires binding to the BtuB receptor for its killing activity, whereas colicin N does not. Clones able to grow when exposed to colicins E2 and E8, but that were still sensitive to E1 are highly unlikely to be *btuB* mutants since E1 requires BtuB for translocation into cells [28,41]. To confirm this, we Sanger sequenced

**Table 2. Colicin resistance phenotyping.**

| Colicin E2 | Colicin E8 | Colicin E1 | Colicin N | Label |
|---|---|---|---|---|
| death | death | death | death | WT |
| growth | death | death | death | E2 immune |
| growth | growth | growth | death | (BtuB) resistant[1] |
| growth | growth | death | death | speculative TolB resistant[2] |

[1]partially sequence-confirmed

[2]not sequence-confirmed

(primers BtuB_BZB.F and BtuB_BZB.R) the *btuB* region of eight resistant clones that emerged during a pilot competition experiment (S6c Fig) and found that 63% (5/8) of clones in the Δ*metE* background had an intact *btuB* (no SNPs detected compared to reference sequence). The three remaining clones had *btuB* mutated by large insertions comparable to those found in the WT background (see above). We speculate that the *btuB*-intact resistant clones might instead have mutations in the gene encoding the periplasmic protein TolB, since it is required for uptake of colicin E2 and E8 but not E1 or N [28,41]. We did not sequence the *tolB* region in any of the tested clones.

**Growth assays on plates.** To confirm vitamin B12-dependence in the engineered Δ*metE* strain, we monitored growth in liquid culture in a plate reader (S6a Fig). Liquid pre-cultures in M9-Glc-B12 of all strains to be tested were inoculated from single colonies grown on LB plates and incubated overnight. For each strain, 1 mL of overnight culture was then washed 4 times in 1 mL M9-Glc (without $B_{12}$), density-adjusted to OD = 1.0, and diluted 1:100 into either LB, M9-Glc, or M9-Glc-B12. Per strain-medium combination, 200 μL of inoculated medium was then added to each of three randomly assigned replicate wells of a 96-well plate (Corning Ref. 3788; clear, round-bottom, polystyrene, not treated; outer wells filled with $dH_2O$). Multi-well plates were incubated for 24 h at 37 °C in a FLUOstar Omega microplate reader (BMG Labtech). OD measurements were taken every 10 min, and plates were shaken (orbital, 400 rpm) for 10 s before every measurement.

**Growth assays on plates.** To compare the growth of strains in mono- and mixed cultures on plates, we conducted growth assays on nutrient agar. Liquid pre-cultures in M9-Glc-B12 of all strains to be tested were inoculated from single colonies grown on LB plates and incubated overnight. For each strain, 1 mL of overnight culture was then washed four times in 1 mL M9-Glc (S6b Fig) or saline (0.85% w/v) (S7a Fig), density-adjusted to OD = 1.0 and diluted 1:100 into either M9-Glc (S6b Fig) saline (0.85% w/v) (S7a Fig). For mixed-strain treatments, strain dilutions were mixed at a 1:1 ratio. For the experiment shown in S6b Fig, three replicate drops of 20 μL per treatment were then spotted on LB, M9-Glc, or M9-Glc-B12 agar. For the experiment shown in S7a Fig, three replicate drops of 50 μL per treatment were spotted on M9-Glc-Srb or M9-Glc-Srb-B12 agar. Droplets were left to dry at room temperature for approximately 30 min before incubating the plates for 24 h at 37 °C. To determine final CFU counts, mixed- and monoculture spots were each resuspended in 1 mL saline (0.85% w/v), serially diluted, and spotted on selective plates as outlined above (see section "Short competitions").

**Data analysis.** Data analysis and visualization were performed using R version 4.2.1 [100] in RStudio version 2023.9.0.463 [101] and packages dplyr [102], Rmisc [103], ggplot2 [104], and cowplot [105]; as well as Matlab version R2023b. For all statistical tests, the significance level *a* was set to 0.05. To test whether target strain survival or transconjugant abundance differed between experiments, unpaired, two-sample *t*-tests on log-transformed CFU counts were performed. To confirm normality of the data, we used Shapiro-Wilk tests on log-transformed CFU counts. To check the equality of variances between samples, we used F tests. In cases where variances between samples were not equal, Welch's *t* test was performed. Sample sizes and statistical details (test values, *p*-values) for each test are provided in the respective figure legends.

**Bioinformatics.** To identify how conserved BtuB is in *E. coli*, 2,601 complete *E. coli* genomes from the BV-BRC database [106] were scanned for the presence of the *btuB* gene using *abricate* [107]. To ensure we are only detecting intact *btuB*, results were filtered to include only those with sequence coverage and identity ≥95% and no gaps in the alignment.

## Supporting information

**S1 Fig. Toxin plasmid transfer in short competitions.** We conducted pairwise competition assays between different *E. coli* strains on LB agar plates. For each genotype, cell recovery (CFU) at each time point of co-culturing is shown. CFU for each time point after *t* = 0 were determined via destructive sampling of *n* = 3 independent replicates (see section "Materials and methods"). Means across replicates are shown as dots and connected by lines. Shaded ribbons around lines depict standard error across replicates. Dashed lines indicate the detection limit (200 CFU). **(a)**

MG1,655-Km$^R$ pColE2-Cm$^R$ ('Attacker') competed against MG1,655-Gm$^R$ ('Target'). No transconjugants (MG1,655-Gm$^R$ pColE2-Cm$^R$) were detected. **(b)** MG1,655-Km$^R$ R751-Sp$^R$ pColE2-Cm$^R$ ('Attacker$^D$') competed against MG1,655-Gm$^R$ ('Target'). 'Transconjugant' (MG1,655-Gm$^R$ R751-Sp$^R$ pColE2-Cm$^R$) CFU are shown as they emerge during the interaction. **(c)** BZB1011-Km$^R$ R751-Sp$^R$ pColE2-oriT-Amp$^R$ ('Attacker$^D$ pColE2-oriT') competed against BZB1011-Cm$^R$ Δ*btuB* ('Target Δ*btuB*'). No transconjugants (BZB1011-Cm$^R$ Δ*btuB* pColE2-oriT-Amp$^R$) were detected ('Transconjugant Δ*btuB*'). **(d)** BZB1011-Km$^R$ R751-Sp$^R$ pColE2-Amp$^R$ ('Attacker$^D$') competed against BZB1011-Cm$^R$ Δ*btuB* R751-Sp$^R$ ('Target Δ*btuB* R751'). Transconjugant (BZB1011-Cm$^R$ Δ*btuB* R751-Sp$^R$ pColE2-Amp$^R$) CFU are shown as they emerge during the interaction ('Transconjugant Δ*btuB* R751'). Data underlying these figures is available from https://doi.org/10.5281/zenodo.10909492.
(TIF)

**S2 Fig. Competitions across a range of initial cell densities.** We conducted pairwise competition assays between different *E. coli* strains on LB agar plates. For each genotype, initial cell density and post-competition cell recovery (CFU) are shown. Competitions were initialized by adjusting preculture cell densities to an optical density (OD) of 0.002, 0.02, 0.2 or 2.0. Means across $n = 3$ replicates are shown as dots and connected by lines. Shaded ribbons around lines depict standard error across replicates. Grey dashed lines indicate the detection limit (200 CFU). Subsets of this dataset are shown in Fig 1d–f. **(a)** BZB1011-Km$^R$ pColE2-Amp$^R$ ('Attacker') competed against BZB1011-Cm$^R$ ('Target'). No transconjugants (BZB1011-Cm$^R$ pColE2-Amp$^R$) were detected. **(b)** BZB1011-Km$^R$ R751-Sp$^R$ pColE2-Amp$^R$ ('Attacker$^D$') competed against BZB1011-Cm$^R$ Δ*btuB* ('Target Δ*btuB*'). Transconjugant (BZB1011-Cm$^R$ Δ*btuB* R751-Sp$^R$ pColE2-Amp$^R$) CFU are shown as they emerge during the interaction ('Transconjugant Δ*btuB*'). **(c)** BZB1011-Km$^R$ R751-Sp$^R$ pColE2-Amp$^R$ ('Attacker$^D$') competed against BZB1011-Cm$^R$ ('Target'). 'Transconjugant' (BZB1011-Cm$^R$ R751-Sp$^R$ pColE2-Amp$^R$) CFU are shown as they emerge during the interaction. Data underlying these figures is available from https://doi.org/10.5281/zenodo.10909492.
(TIF)

**S3 Fig. Niche separation influences strain frequencies in competitions with killing and transfer.** Using a constant initial pool of nutrients ($N_1 + N_2 + N_3 = 3.0$), we observe the impact of varying levels of niche separation (i.e., metabolic diversity) on final strain frequencies. **(a)** In a scenario with both killing and transfer, the final frequency of the transconjugants is lowest when niche separation is lowest (left side of plot; $N_1 = 3.0$; $N_2 + N_3 = 0$). As niche separation increases ($N_1$ = decreasing; $N_2 + N_3$ = increasing), the final frequency of transconjugants increases and the final frequency of attackers decreases. Maximum transconjugant frequency is observed with complete niche separation (right side of plot; $N_1 = 0$; $N_2 = N_3 = 1.5$). **(b)** In a scenario with only killing, attackers dominate across all conditions. **(c)** In a scenario with only transfer, final strain frequencies are constant regardless of niche separation. Across all conditions, $N_2 = N_3$. All parameters are default (Table 1) unless stated. Code and data underlying these figures are available from https://doi.org/10.5281/zenodo.14910561.
(TIF)

**S4 Fig. Competitions involving metabolic diversity across a range of initial cell densities.** We conducted pairwise competition assays between different *E. coli* strains on LB agar plates. For each genotype, initial cell density and post-competition cell recovery (CFU) are shown. Competitions were initialized by adjusting preculture cell densities to an optical density (OD) of 0.002, 0.02, 0.2 or 2.0. Means across $n = 3$ replicates are shown as dots and connected by lines. Shaded ribbons around lines depict standard error across replicates. Grey dashed lines indicate the detection limit (200 CFU). Subsets of this dataset are shown in Fig 4a–c. **(a)** BZB1011-Km$^R$ Δ*srlAEB* ('Δ*srlAEB*') competed against BZB1011-Cm$^R$ ('WT'). **(b)** BZB1011-Km$^R$ Δ*srlAEB* pColE2-Amp$^R$ ('Attacker Δ*srlAEB*') competed against BZB1011-Cm$^R$ ('Target'). No transconjugants (BZB1011-Cm$^R$ pColE2-Amp$^R$) were detected. **(c)** BZB1011-Km$^R$ Δ*srlAEB* R751-Sp$^R$ pColE2-Amp$^R$ ('Attacker$^D$ Δ*srlAEB*') competed against BZB1011-Cm$^R$ Δ*btuB* ('Target Δ*btuB*'). Transconjugant

(BZB1011-Cm$^R$ Δ*btuB* R751-Sp$^R$ pColE2-Amp$^R$) CFU are shown as they emerge during the interaction ('Transconjugant Δ*btuB*'). **(d)** BZB1011-Km$^R$ Δ*srlAEB* R751-Sp$^R$ pColE2-Amp$^R$ ('Attacker$^D$ ΔsrlAEB') competed against BZB1011-Cm$^R$ ('Target'). 'Transconjugant' (BZB1011-Cm$^R$ R751-Sp$^R$ pColE2-Amp$^R$) CFU are shown as they emerge during the interaction. Data underlying these figures is available from https://doi.org/10.5281/zenodo.10909492. (TIF)

**S5 Fig. A two-nutrient model of metabolic diversity also favors transconjugants.** Modeling scenarios for each column are shown across the top row. **(a–c)** Example dynamics of the strains (attacker, target, transconjugant) during a contest using parameters that corresponding to the cross (X) shown in the parameter sweeps directly below **(d–f)** Transconjugant frequency at steady state (see section "Materials and methods") in competitions as a function of conjugation rate (*b*) and toxin killing efficiency (*E*). **(g–i)** Final frequency of different strain types at steady state as a function of initial frequency of target and attacker strains. In the two-nutrient model, $N_3 = 0.25$ in order to keep maximum observed growth rates similar for all strains. All other parameters are default (Table 1) unless stated. Code and data underlying these figures are available from https://doi.org/10.5281/zenodo.14910561. (TIF)

**S6 Fig. Characterization of the Δ*metE* mutant in minimal medium. (a)** Growth curves of BZB1011-Cm$^R$ ('WT Cm$^{R'}$') and Δ*metE* ('ΔmetE Cm$^{R'}$') in different nutrient media (LB, M9-Glc and M9-Glc-B12). Mean OD across *n* = 3 replicates are depicted as dots and connected by lines. Shaded ribbons around lines represent standard error across replicates. **(b)** BZB1011-Km$^R$ ('WT Km$^{R'}$') and BZB1011-Cm$^R$ Δ*metE* ('ΔmetE Cm$^{R'}$') were grown for 24 h in either mono- or mixed cultures on different nutrient medium agar plates (LB, M9-Glc or M9-Glc-B12). Dotted line indicates initial cell density for all strains. Final CFU for *n* = 3 replicates and their means are shown as faint and solid color dots, respectively. **(c–e)** Pairwise competition assays on minimal medium agar plates. Initial cell density and post-competition cell recovery (CFU) for each genotype are shown. Means across *n* = 3 independent replicates are depicted as dots and connected by lines. For 'Target' data depicted in panel d (top left), *n* = 2. Shaded ribbons around lines depict standard error across replicates. Grey dashed lines indicate the detection limit (200 CFU). To test for differences in target survival, we used two-sided, two-sample *t*-tests on log-transformed CFU counts. (c; left) versus (c; right): *t* = −10.05, *df* = 4, *p* < 0.001. (e; top left) versus (e; top right): *t* = −22.64, *df* = 4, *p* < 0.001. **(c+d)** BZB1011-Km$^R$ pColE2-Amp$^R$ ('Attacker') competed against either BZB1011-Cm$^R$ or BZB1011-Cm$^R$ Δ*metE* as 'Target'. Panel **c** shows data from a pilot experiment used for sequencing spontaneously resistant clones (see section "Materials and methods"). **(e)** BZB1011-Km$^R$ R751-Sp$^R$ pColE2-Amp$^R$ ('Attacker$^D$') competed against BZB1011-Cm$^R$ (Target). 'Transconjugant' (BZB1011-Cm$^R$ (Δ*metE*) R751-Sp$^R$ pColE2-Amp$^R$) CFU are shown as they emerge during the interaction. Data underlying these figures is available from https://doi.org/10.5281/zenodo.10909492. (TIF)

**S7 Fig. Characterization of the Δ*metE* mutant in minimal medium supplemented with sorbitol. (a)** BZB1011-Km$^R$ ('WT Km$^{R'}$') and BZB1011-Cm$^R$ Δ*metE* ('ΔmetE Cm$^{R'}$') were grown for 24 h in mono- and mixed cultures on minimal medium agar plates supplemented with glucose and sorbitol, either with (M9-Glc-Srb-B12) or without vitamin B12 (M9-Glc-Srb). Dotted line indicates initial cell density for all strains. Final CFU for *n* = 3 replicates and their means are shown as faint and solid color dots, respectively. **(b–d)** BZB1011-Km$^R$ Δ*srlAEB* ('ΔsrlAEB Km$^{R'}$') was competed either against (b, c) BZB1011-Cm$^R$ ('WT Cm$^{R'}$'); or (d) BZB1011-Cm$^R$ Δ*metE* ('ΔmetE Cm$^{R'}$') for 24 h on minimal medium agar plates, with or without the addition of sorbitol. For each strain, initial cell density and post-competition cell recovery (CFU) are shown. Means across *n* = 3 replicates are depicted as dots and connected by lines. Shaded ribbons around lines represent standard error across replicates. Grey dashed lines indicate the detection limit (200 CFU). Glucose concentrations in minimal medium were either the standard 0.2% (b) or were experimentally varied from 0.05% to

0.4% (c and d). Yellow highlight indicates standard media conditions used in experiments shown in <u>Fig 5</u> and (a), (b), (e) and (f). **(e)** BZB1011-Km$^R$ $\Delta srlAEB$ R751-Sp$^R$ pColE2-Amp$^R$ ('$\Delta$srlAEB R751 pColE2) was competed either against BZB1011-Cm$^R$ R751-Sp$^R$ pColE2-Amp$^R$ or against BZB1011-Cm$^R$ $\Delta metE$ R751-Sp$^R$ pColE2-Amp$^R$ for 24 h on minimal medium agar supplemented with sorbitol and vitamin B12 (M9-Glc-Srb-B12). For each strain, initial cell density and post-competition cell recovery (CFU) are shown. Means across $n$ = 3 replicates are depicted as dots and connected by lines. Shaded ribbons around lines represent standard error across replicates. Grey dashed lines indicate the detection limit (200 CFU). **(f)** BZB1011-Cm$^R$ ('WT') and BZB1011-Cm$^R$ $\Delta metE$ ('metE'), either with or without carrying plasmids R751-Sp$^R$ and pColE2-Amp$^R$, were serially passaged in monocultures over 3 days. CFU were determined each day before transferring to fresh nutrient plates (see section "Materials and methods"). Means across $n$ = 3 biological replicates are connected by lines. Shaded ribbons around lines represent standard error across replicates. Grey dashed lines indicate the detection limit (200 CFU). Data underlying these figures is available from <u>https://doi.org/10.5281/zenodo.10909492</u>.
(TIF)

**S8 Fig. Final frequencies of transconjugant (a and b), attacker (c and d), and target (e and f) strains across variable initial population densities and strain frequency.** Each heat map ranges from initial population density of 0.01 to 1.0, and varies the initial frequency of the target and attacker strains from 99:1 to 1:99. Scenarios of complete niche overlap **(a, c, e)** and metabolic diversity **(b, d, f)** are represented. All parameters are default unless otherwise noted. Code and data underlying these figures are available from <u>https://doi.org/10.5281/zenodo.14910561</u>.
(TIF)

**S9 Fig. Final frequency of transconjugants based on varying initial parameter values.** One parameter value changed at a time, while all others are held constant (default values). Modeling scenario with complete niche overlap represented on left side of each pair of plots (side by side), with the metabolic diversity scenario represented on the right side. Code and data underlying these figures are available from <u>https://doi.org/10.5281/zenodo.14910561</u>.
(TIF)

**S1 Table. Resistance phenotypes.**
(DOCX)

**S2 Table. Strains and plasmids used in this study.**
(DOCX)

**S3 Table. Primers used in this study.**
(DOCX)

## Acknowledgements

We thank James Wilson for plasmids, and Erik Bakkeren and Sean Booth for strains and plasmids.

## Author contributions

**Conceptualization:** Elisa T. Granato, Kevin R. Foster.

**Formal analysis:** Elisa T. Granato.

**Funding acquisition:** Elisa T. Granato, Kevin R. Foster.

**Investigation:** Elisa T. Granato, Christian Kirk, Connor Sharp.

**Methodology:** Elisa T. Granato, Jacob D. Palmer, George Shillcock.

**Resources:** Kevin R. Foster.

**Software:** Jacob D. Palmer, George Shillcock.

**Supervision:** Elisa T. Granato, Jacob D. Palmer, Kevin R. Foster.

**Visualization:** Elisa T. Granato, Jacob D. Palmer.

**Writing – original draft:** Elisa T. Granato, Jacob D. Palmer.

**Writing – review & editing:** Elisa T. Granato, Jacob D. Palmer, Christian Kirk, Connor Sharp, George Shillcock, Kevin R. Foster.

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
