## [Editor Report · Decision Letter 0]

9 Sep 2024

Dear Dr Granato, 

Thank you for submitting your manuscript entitled "Horizontal gene transfer can reshape bacterial warfare" for consideration as a Research Article by PLOS Biology.

Your manuscript has now been evaluated by the PLOS Biology editorial staff, as well as by an academic editor with relevant expertise, and I am writing to let you know that we would like to send your submission out for external peer review.

Once your full submission is complete, your paper will undergo a series of checks in preparation for peer review. After your manuscript has passed the checks it will be sent out for review. To provide the metadata for your submission, please Login to Editorial Manager (https://www.editorialmanager.com/pbiology) within two working days, i.e. by Sep 11 2024 11:59PM.

Kind regards,

Melissa

Melissa Vazquez Hernandez, Ph.D.

Associate Editor

PLOS Biology

---

## [Decision Letter · Decision Letter 1]

14 Oct 2024

Dear Elisa,

Thank you for your patience while your manuscript "Horizontal gene transfer can reshape bacterial warfare" was peer-reviewed at PLOS Biology. I have taken over its handling in the absence colleague Melissa from the office this week, to prevent unnecessary loss of time. Your study has now been evaluated by the PLOS Biology editors, an Academic Editor with relevant expertise, and by three independent reviewers, the first one of which (Ellie Harrison) has signed her report. 

In light of the reviews, which you will find at the end of this email, we would like to invite you to revise the work to thoroughly address the reviewers' reports.

As you will see below, the reviewers are all supportive of the work, but R2 and R3 consider that a revision strengthening the modelling aspects of the work is needed before we can consider it for publication. The revision would need to improve the exploration of parameters in the model, and ideally address the issue of environmental structure (especially considering your expertise on the subject). We consider that addressing the community composition aspect is beyond the scope of work, but it would need to be discussed explicitly. Given the nature of the revision needed, we cannot make a decision about publication until we have seen the revised manuscript and your response to the reviewers' comments. Your revised manuscript is likely to be sent for further evaluation by a subset of the reviewers.

**IMPORTANT - SUBMITTING YOUR REVISION**

*Re-submission Checklist*

*Published Peer Review*

*PLOS Data Policy*

*Blot and Gel Data Policy*

Sincerely,

Nonia

Nonia Pariente, PhD

Editor in Chief

PLOS Biology 

on behalf of 

Melissa

Melissa Vazquez Hernandez, Ph.D.

Associate Editor

PLOS Biology

REVIEWS:

Reviewer #1: Ellie Harrison

Reviewer #1: This is a really nicely implemented dissection of the impact of mobility on bacterial 'weapons' involved in competition. The authors combine modelling and experiments to break down the impacts of gene mobility for competition in a really clear and readable way. They show clearly that the hypothesis that HGT to the target of competition will not always reduce the effectiveness of the competition (which is what we often see for temperate phages where transmission is basically inherent to attack and competition often flames out quickly). They then show convincingly that in more nuanced and realistic communities the effect of transfer can have greater effects. 

I commend the authors here on their excellent work and have only one minor, somewhat philosophical point to make, which may or may not augment the discussion. As they state their work nicely supports the idea that the real benefactor of transfer of these gene cassettes is the mobile element. However the search for conditions which allow the recipients to out complete the donors might be less important when seen from the plasmid's eye view. Either way the plasmid wins in these short term competitions - but in the long term the bet hedging (I agree) is key. Therefore i would assume that the resource heterogeneity principle also works in a temporarily or spatially heterogenous environment. 

Reviewer #2: 

The manuscript "Horizontal gene transfer can reshape bacterial warfare" by Granato et al. addresses a highly significant and thought-provoking question. Bacteria frequently encode toxins, such as bacteriocins. Cells harboring genes for these toxins gain a distinct advantage by eliminating nearby sensitive cells. However, a paradox arises: why are toxin genes predominantly located on conjugative or mobilizable plasmids rather than on the bacterial chromosome? This presents a conundrum, as plasmids carry the toxin and the corresponding immunity genes, which transfer to neighboring cells, rendering those cells immune to the toxin's effects. 

The question posed is highly relevant. As the authors mention in the Discussion, in Escherichia coli, all colicins are plasmid-encoded rather than chromosomally encoded, a phenomenon for which no explanation has been provided until now. This issue is of particular significance to Microbiology, where toxins are prevalent, and Evolutionary Biology, where genetic conflicts play a central role. Furthermore, in the context of health, this manuscript offers valuable insights into the optimal strategy for enhancing the success of toxin genes within a microbiome (for example).

The authors propose the following explanatory hypothesis: encoding toxin/immunity genes on plasmids (rather than on the chromosome) provides an advantage to these genes when the donor and potential recipient cells do not compete intensely for nutrients. The authors employ mathematical and computational modeling and experiments to test this hypothesis.

A) The manuscript is very clear, except for the Abstract. In my (modest) opinion, the abstract doesn't do justice to the aim (scientific question) and the work done. For example, the second sentence of the abstract is:"Genes encoding many of these weapons can be transmitted horizontally, but the impact on bacterial warfare is not understood.". What do you mean? Why is it not understood? What is missing? Consider, for example, these two sentences: "The horizontal mobility of toxin and immunity genes (…) presents a unique problem. If gene transfer happens during an encounter with a competitor, the target becomes immune to the toxin, rendering the bacterial weapon useless and negating any potential fitness benefits for the producer. ". These are very clear sentences in the Introduction (Lines 39-41): much clearer than those of the abstract because they clearly state the scientific problem. My suggestion is to write something like this in the Abstract.

B) The experiments performed corroborate the hypothesis. However, contrary to the authors' claim, the computer/mathematical model could not "predict" the experimental results. Why? Because the authors made a model of a liquid (non-structured) environment, whereas the experiments were in plates. Interestingly, it is precisely with colicins that Lin Chao and Bruce Levin (1981, PNAS) have shown a strong difference between the two environments (regarding expected experimental outcomes). I do not think this is a problem for this manuscript. Still, I ask the authors to state this point clearly. That is, the verbal hypothesis is corroborated in the "liquid" simulations ("liquid" because they used ordinary differential equations) and in a structured environment. It should be clear that the author's aim is not to prove that the results of the experiments corroborate the computer model. It is to show that their hypothesis is corroborated by simulations of an unstructured environment and experiments in a structured environment. I suggest small changes in the text. For example, in line 101, one can read, "To test the predictions of our model empirically, we turned to a well-characterized family of antibacterial toxins: the colicins."; I suggest changing this to something like "To test our hypothesis empirically, we turned to a well-characterized family of antibacterial toxins: the colicins." 

C) In lines 233 - 235, one reads the following: "Note that this experimental design is slightly different to our second model, because only one of the two strains has a private nutrient, rather than both as in our modelling. However, we show in the supplement that this change does not affect our modelling predictions (Fig. S5)". I understand why it is equivalent, but why not do the modeling in that way: only one of the two strains has a private nutrient? 

D) Statistics: The authors should apply Shapiro-Wilk tests (or others) to ensure the normality of the data before performing the t-test (or use a non-parametric test if the data are not normal). 

E) Statistical tests appear mostly at the end of each figure legend. I would prefer to read the statistics in the main text. 

F) In lines 242-243, the authors tested the difference between fig 2f and fig 4c, but the corresponding result is "hidden" at the bottom of the legend of figure 4. By the way, at the bottom of the legend of figure 4 should be "Fig. 2f and Fig. 4c", not "Fig. 2c and Fig. 4f".

G) The experiments are very clear. If possible, I would like to know better the systems used in the following sense: What is the Inc group of the plasmid encoding the toxin? Also, do the authors know what the helper plasmid is? If the authors have these data, including this information in the Methods section would be useful.

H) Figures are fine, but Figs 1(d-f) or 3(d-f), etc, are not very clear. For example, in Figs 3e or 3f, it is unclear how green they are (is it closer to 0.5 or 1?).

Minor points:

1 - Lines 32-34: I have some problems with the sentence "Genes encoding the production of bacteriocins, antibiotics, and even complex weapons like the contractile type VI secretion machinery, can be horizontally transmitted between cells via transformation (6), conjugation (7-15) and transduction (16,17).". Genes encoding "everything" can be horizontally transmitted between cells via transformation, conjugation and transduction. This is not something special about toxin/immunity genes. I suggest to the authors to change this sentence.

2- Line 245 phrasing: "abundances OF the attacker"?

3 - Line 274 "Collectively, these results support our model's prediction that access to a private nutrient can empower transconjugants and promote their invasion into a population of toxin plasmid donors.". Could the authors clarify the meaning of "invasion" here, or clarify the sentence?

4- Line 365: delete the "," after "could not use"

5- Line 368-369: I did not understand this sentence.

6-Lines 416-8: rephrase

7- There is a problem with Table S1 (formatting)

Reviewer #3: 

This manuscript describes a study to test the hypothesis that conjugation of a plasmid carrying a toxin may change the dynamics of antagonistic interactions between two bacterial populations that are dominated by the antagonism associated with the focal toxin. First, a model assesses the hypothesis relative to a scenario without transfer and a scenario where there is transfer but no toxin. Second, the hypothesis is tested experimentally, showing that conjugation makes relatively little impact in the dynamics. A more complex model allowing for different niches for the bacteria (specific sources of nutrients) suggests an increased impact of the transfer, which is confirmed experimentally. This suggests that horizontal gene transfer of toxins (which is frequent) can impact competition at the ecological scale, which is an interesting and important result. The text is clear and figures excellent (even if legends could be improved). 

Major comments.

While the experimental part is very much developed, the modeling part seems insufficient to sustain some of the statements. There are several issues. 

- The model is a standard differential equations model, but experiences are done in a somewhat structured environment, which may affect its applicability. 

- There are several claims in that the model shows some events are too rare or too frequent. However, there seems to be no clear biological parametrization of the model, i.e. the parameters are not reflecting real conjugation rates, real costs of producing the toxin, etc. Under these conditions, the model can't produce quantitatively relevant predictions (only qualitative). 

- A more thorough exploration of the results of the model when varying the parameters seems essential, especially given that parameters were not obtained from experimental data.

- The model seems to start with very small populations that grow to stationary phase. If the goal is to mimic experimental conditions, this is fine. But in real life the populations of residents are probably high and there may be a steady state of growth. It is not clear to me that the results for the invaders stand when residents are both frequent and initially abundant. 

One point that should be mentioned, maybe even modeled given the team's experience in the field, is the impact of environmental structure (that can dramatically change this type of dynamics). Of note, these plasmids cannot conjugate in liquid medium. The efficiency of the toxin (or at least its impact on the growth kinetics) may also vary depending on environmental structure.

Another point, which in this case may be too complex to add to models and experiences but would be important to discuss, relates with community composition. Many recent works about antibiotic resistance or resistance to phages have shown that results obtained in models or experiences using bacterial communities reveal completely different dynamics from those using clonal minimal systems (as done here). There are several arguments that could apply to this study: the cost of resistance (discussed here in another context), the cost of production (in face of competitors that may be immune since colicins have narrow ranges), the dilution effect that may affect both conjugation (briefly discussed) and killing. 

Minor comments

-While figures are very clear, some legends are not. They are long, confusing and they don't always clearly describe each panel. They deserve editing. For example legend of Figure 2. 

-Line 33. I suggest rephrasing. The sentence suggests that type VI secretion systems can be transferred by transformation or transduction, which has not been shown to the best of my knowledge (the cited references are about effectors not the system).

-Line 104. Some colicins are chromosomal (Poey, AAC, 2006)

-Line 373. Low density of E coli may affect conjugation, but also killing by the colicin, which may be neutralized in non-susceptible bacteria? 

-Line 397. See comment above about colicins in chromosomes. Also, it's not clear what is the intent of this statement. Why would E. coli be different from the others? And in the chromosomes, aren't the bacteriocins also in MGEs like ICE? 

-Line 414 and next. This part of the text is confusing and deserves re-writing. 

-Line 462. This is incorrect, the literature does not demonstrate that conjugative plasmids always transfer at much higher rates than cognate mobilizable elements. Several examples have been published of similar or close rates (see Charneco, Mic Biotech, 24 or Gaissmaier, Jbac, 22 for recent ones). In some cases the rates of mobilization of the mobilizable element are much higher than those of the conjugative (see Murányi, NAR, 24). Note that many experimental works in the field use (the same) extremely active conjugative plasmids, which creates a bias. (there may also be a confusion here between mobilization and facilitation, as defined in reference 88)

-It would help to have units in the parameters of Table 1. 

-In my pdf of table S1 there are several "Error! Bookmark not defined." needing correction

---

## [Editor Report · Decision Letter 2]

12 Feb 2025

Dear Dr Granato,

Thank you for your patience while we considered your revised manuscript "Horizontal gene transfer can reshape bacterial warfare" for publication as a Research Article at PLOS Biology. This revised version of your manuscript has been evaluated by the PLOS Biology editors, and the Academic Editor.

Based on our Academic Editor's assessment of your revision, we are likely to accept this manuscript for publication, provided you satisfactorily address the remaining editorial points. Please also make sure to address the following data and other policy-related requests.

a) We routinely suggest changes to titles to ensure maximum accessibility for a broad, non-specialist readership, and to ensure they reflect the contents of the paper. Please ensure you change both the manuscript file and the online submission system, as they need to match for final acceptance:

"Horizontal gene transfer of molecular weapons can reshape interbacterial competition"

b) Please add the supplementary references to the references in the main text

Please supply the numerical values either in the a supplementary file or as a permanent DOI’d deposition for the following figures:

Figure 1a-i, 2a-f, 3a-i, 4a-e, 5a-d, S1a-d, S2a-c, S4a-d, S5a-i, S6a-e, S7a-f, S8a-f, S9a-f

d) Please provide the link to Zenodo of all the datasets generated

e) Thank you for mentioning that the code will be available on Github. However, because Github depositions can be readily changed or deleted, please make a permanent DOI’d copy (e.g. in Zenodo) and provide this URL in the manuscript and Data Availability Statement.

We expect to receive your revised manuscript within two weeks. 

*Published Peer Review History*

*Press*

Sincerely,

Melissa

Melissa Vazquez Hernandez, Ph.D.

Associate Editor

PLOS Biology

---

## [Editor Report · Decision Letter 3]

4 Mar 2025

Dear Dr Granato,

Thank you for the submission of your revised Research Article "Horizontal gene transfer of molecular weapons can reshape bacterial competition" for publication in PLOS Biology. On behalf of my colleagues and the Academic Editor, Sam P. Brown, I am pleased to say that we can in principle accept your manuscript for publication, provided you address any remaining formatting and reporting issues. These will be detailed in an email you should receive within 2-3 business days from our colleagues in the journal operations team; no action is required from you until then. Please note that we will not be able to formally accept your manuscript and schedule it for publication until you have completed any requested changes.

PRESS

Sincerely, 

Melissa

Melissa Vazquez Hernandez, Ph.D., Ph.D.

Associate Editor

PLOS Biology
